# CHIP: CHannel Independence-based Pruning for Compact Neural Networks

**Yang Sui   Miao Yin   Yi Xie   Huy Phan   Saman Zonouz   Bo Yuan**
Department of Electrical and Computer Engineering
Rutgers University
Piscataway, NJ 08854, USA
{yang.sui, miao.yin, yi.xie, huy.phan, saman.zonouz}@rutgers.edu,
bo.yuan@soe.rutgers.edu

## Abstract

Filter pruning has been widely used for neural network compression because of its enabled practical acceleration. To date, most of the existing filter pruning works explore the importance of filters via using intra-channel information. In this paper, starting from an inter-channel perspective, we propose to perform efficient filter pruning using *channel independence*, a metric that measures the correlations among different feature maps. The less independent feature map is interpreted as containing less useful information/knowledge, and hence its corresponding filter can be pruned without affecting model capacity. We systematically investigate the quantification metric, measuring scheme and sensitiveness/reliability of channel independence in the context of filter pruning. Our evaluation results for different models on various datasets show the superior performance of our approach. Notably, on CIFAR-10 dataset our solution can bring $0.90\%$ and $0.94\%$ accuracy increase over baseline ResNet-56 and ResNet-110 models, respectively, and meanwhile the model size and FLOPs are reduced by $42.8\%$ and $47.4\%$ (for ResNet-56) and $48.3\%$ and $52.1\%$ (for ResNet-110), respectively. On ImageNet dataset, our approach can achieve $40.8\%$ and $44.8\%$ storage and computation reductions, respectively, with $0.15\%$ accuracy increase over the baseline ResNet-50 model. The code is available at https://github.com/Eclipsess/CHIP_NeurIPS2021.

## 1   Introduction

Convolutional neural networks (CNNs) have obtained widespread adoptions in numerous important AI applications [17, 48, 47, 12, 11, 44, 35]. However, CNNs are inherently computation intensive and storage intensive, thereby posing severe challenges for their efficient deployment on resource-constrained embedded platforms. To address these challenges, *model compression* is widely used to accelerate and compress CNN models on edge devices. To date, various types of compression strategies, such as network pruning [15, 16, 37, 60, 28, 14, 1, 49, 10, 63, 36, 18, 13, 3, 2, 25, 53, 50, 9, 39, 33], quantization [15, 55, 43, 8], low-rank approximation [56, 40, 58, 57], knowledge distillation [22, 41] and structured matrix-based construction [45, 29, 6], have been proposed and explored. Among them, *network pruning* is the most popular and extensively studied model compression technique in both academia and industry.

Based on their differences in pruning granularity, pruning approaches can be roughly categorized to *weight pruning* [16, 15] and *filter pruning* [54, 27, 21, 38, 34]. Weight pruning focuses on the proper selection of the to-be-pruned weights within the filters. Although enabling a high compression ratio, this strategy meanwhile causes unstructured sparsity patterns, which are not well supported by the general-purpose hardware in practice. On the other hand, filter pruning emphasizes the removal of

35th Conference on Neural Information Processing Systems (NeurIPS 2021).

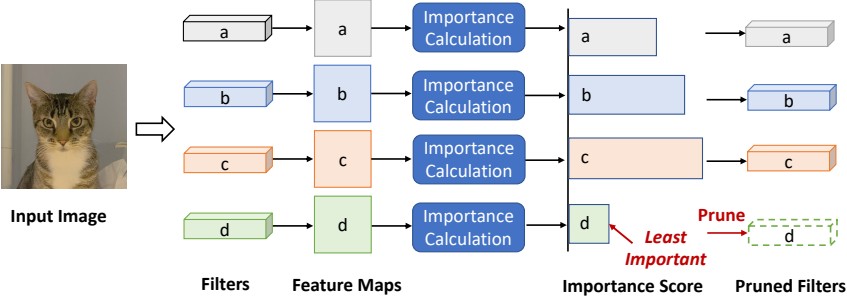

(a) Feature information-based filter pruning from an intra-channel perspective.

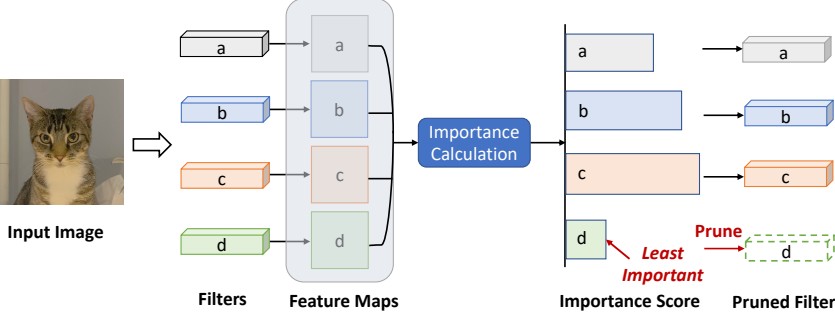

(b) Feature information-based filter pruning from an inter-channel perspective.

Figure 1: Intra-channel vs Inter-channel perspectives for filter pruning.

the entire selected filters. The resulting structured sparsity patterns can be then properly leveraged by the off-the-shelf CPUs/GPUs to achieve acceleration in the real-world scenario.

**Existing Filter Pruning Methods.** Motivated by the potential practical speedup offered by filter pruning, to date numerous research efforts have been conducted to study *how to determine the important filters* – the key component of efficient filter pruning. A well-known strategy is to utilize the norms of different filters to evaluate their importance. Such a "smaller-norm-less-important" hypothesis is adopted in several pioneering filter pruning works [27, 19]. Later, considering the limitations of norm-based criterion in real scenarios, [20] proposes to utilize geometric median-based criterion. More recently, first determining those important feature maps and then preserving the corresponding filters, instead of directly selecting the filters, become a popular strategy for filter pruning. As indicated in [31], the features, by their natures, reflect and capture rich and important information and characteristics of both input data and filters, and hence measuring the importance of features can provide a better guideline to determine the important filters. Built on this pruning philosophy, several feature-guided filter pruning approaches [31, 51] have been proposed and developed, and the evaluation results show their superior performance over the state-of-the-art filter-guided counterparts with respect to both task performance (e.g., accuracy) and compression performance (e.g., model size and floating-point operations (FLOPs) reductions).

**Determining Importance: Intra-channel & Inter-channel Perspectives.** These recent advancements on filter pruning indeed show the huge benefits of leveraging feature information to determine the importance of filters. To date, some feature-guided approaches measure the importance from the *intra-channel* perspective. In other words, no matter which importance metric is used, the importance of one feature map (and its corresponding filter), is measured only upon the information of this feature map in its own channel. On the other aspect, the *inter-channel perspective*, which essentially determines the filter importance via using cross-channel information [20, 42, 46, 52, 26], is still being further explored. To be specific, [20] and [42] adopt cross-channel geometric median and Hessian, respectively, to measure the channel importance. However, such measurement is based on filter instead of feature map information, and hence the rich and important feature characteristics are not properly identified and extracted. [52, 26] also explore inter-channel-based filter pruning via introducing budget constraints across channels. However, such exploration and utilization of the inter-channel information are implicit and indirect, thereby limiting the practical pruning performance.

**Benefits of Inter-channel Perspective.** In principle, the feature information across multiple channels, if being leveraged properly, can potentially provide richer knowledge for filter pruning than the intra-channel information. Specifically, this is because: 1) the importance of one filter, if being solely determined by its corresponding feature map, may be sensitive to input data; while the cross-channel feature information can bring more stable and reliable measurement; and 2) consider the essential mission of pruning is to remove the unnecessary redundancy, the inter-channel strategy can inherently better identify and capture the potential unnecessary correlations among different feature maps (and the corresponding filters), and thereby unlocking the new opportunity of achieving better task and compression performance.

**Technical Preview and Contributions.** Motivated by these promising potential benefits, in this paper we propose to explore and leverage the cross-channel feature information for efficient filter pruning. To be specific, we propose ***Channel Independence***, a cross-channel correlation-based metric to measure the importance of filters. Channel independence can be intuitively understood as the measurement of "replaceability": when the feature map of one filter is measured as exhibiting lower independence, it means this feature map tends to be more linearly dependent on other feature maps of other channels. In such a scenario, the contained information of this low-independence feature map is believed to have already been implicitly encoded in other feature maps – in other words, it does not contain useful information or knowledge. Therefore the corresponding filter, which outputs this low-independence feature map, is viewed as unimportant and can be safely removed without affecting the model capacity. Overall, the contributions of this paper are summarized as:

- We propose channel independence, a metric that measures the correlation of multiple feature maps, to determine the importance of filters. Built from an inter-channel perspective, channel independence can identify and capture the filter importance in a more global and precise way, thereby providing a better guideline for filter pruning.

- We systematically investigate and analyze the suitable quantification metric, the complexity of the measuring scheme and the sensitiveness & reliability of channel independence, and then we develop a low-cost fine-grained high-robustness channel independence calculation scheme for efficient filter pruning.

- We empirically apply the channel independence-based importance determination in different filter pruning tasks. The evaluation results show that our proposed approach brings very high pruning performance with preserving high accuracy. Notably, on CIFAR-10 dataset our solution can bring $0.90\%$ and $0.94\%$ accuracy increase over baseline ResNet-56 and ResNet-110 models, respectively, and meanwhile the model size and FLOPs are reduced by $42.8\%$ and $47.4\%$ (for ResNet-56) and $48.3\%$ and $52.1\%$ (for ResNet-110), respectively. On ImageNet dataset, our approach can achieve $40.8\%$ and $44.8\%$ storage and computation reductions, respectively, with $0.15\%$ accuracy increase over the baseline ResNet-50 model.

## 2 Preliminaries

**Filter Pruning.** For a CNN model with $L$ layers, its $l$-th convolutional layer $\mathcal{W}^l = \{\mathcal{F}_1^l, \mathcal{F}_2^l, \cdots, \mathcal{F}_{c^l}^l\}$ contains $c^l$ filters $\mathcal{F}_i^l \in \mathbb{R}^{c^{l-1} \times k^l \times k^l}$, where $c^l$, $c^{l-1}$ and $k^l$ denote the number of output channels, the number of input channels and the kernel size, respectively. In general, network pruning can be formulated as the following optimization problem:

$$\min_{\{\mathcal{W}^l\}_{l=1}^L} \mathcal{L}(\mathcal{Y}, f(\mathcal{X}, \mathcal{W}^l)), \text{s.t.} \|\mathcal{W}^l\|_0 \leq \kappa^l, \tag{1}$$

where $\mathcal{L}(\cdot, \cdot)$ is the loss function, $\mathcal{Y}$ is the ground-truth labels, $\mathcal{X}$ is the input data, and $f(\cdot, \cdot)$ is the output function of CNN model $\{\mathcal{W}^l\}_{l=1}^L$. Besides, $\|\cdot\|_0$ is the $\ell_0$-norm that measures the number of non-zero filters in the set, and $\kappa^l$ is the number of filters to be preserved in the $l$-th layer.

**Feature-guided Filter Pruning.** Consider the feature maps, in principle, contain rich and important information of both filters and input data, approaches using feature information have become popular and achieved the state-of-the-art performance for filter pruning. To be specific, unlike the filter-guided methods that directly minimize the loss function involved with filters (as Eq. 1), the objective of feature-guided filter pruning is to minimize the following loss function:

$$\min_{\{\mathcal{A}^l\}_{l=1}^L} \quad \mathcal{L}(\mathcal{Y}, \mathcal{A}^l), \text{s.t.} \quad \|\mathcal{A}^l\|_0 \leq \kappa^l, \tag{2}$$

where $\mathcal{A}^l = \{A_1^l, A_2^l, \cdots, A_{c^l}^l\} \in \mathbb{R}^{c^l \times h \times w}$ is a set of feature maps output from the $l$-th layer, and $A_i^l \in \mathbb{R}^{h \times w}$ is the feature map corresponds to the $i$-th channel. In general, after the $\kappa^l$ important feature maps are identified and selected, their corresponding $\kappa^l$ filters are preserved after pruning.

## 3 The Proposed Method

### 3.1 Motivation

As formulated in Eq. 2, feature-guided filter pruning leverages the generated feature maps in each layer to identify the important filters. To achieve that, various types of feature information, such as the high ranks [31] and the scaling factors [51], have been proposed and utilized to select the proper feature maps and the corresponding filters. A common point for these state-of-the-art approaches is that all of them focus on measuring the importance via using the information contained in each feature map. On the other hand, the correlation among different feature maps, as another type of rich information provided by the neural networks, is little exploited in the existing filter pruning works.

**Why Inter-channel Perspective?** We argue that the feature information across multiple channels is of significant importance and richness, and it can be leveraged towards efficient filter pruning. Such an inter-channel perspective is motivated by two promising benefits. First, filter pruning is essentially a data-driven strategy. When the importance of one filter solely depends on the information represented by its own generated feature map, the measurement of the importance may be unstable and sensitive to the slight change of input data. On the other hand, determining the importance built upon information contained in the multiple feature maps, if performed properly, can reduce the potential disturbance incurred by the change of input data, and thereby making the importance ranking more reliable and stable. Second, the inter-channel strategy, by its nature, can better model and capture the cross-channel correlation. In the context of model compression, these identified correlations can be interpreted as a type of architecture-level redundancy, which is exactly what filter pruning aims to remove. Therefore, inter-channel strategy can enable more aggressive pruning while still preserving high accuracy.

### 3.2 Channel Independence: A New Lens for Filter Importance

**Key Idea.** Motivated by these promising benefits, we propose to explore the filter importance from the inter-channel perspective. Our key idea is to use *channel independence* to represent the importance of each feature map (and its corresponding filter). To be specific, when one feature map of one channel is highly *linearly dependent* on other feature maps of other channels, it implies that its contained information has already been largely encoded in other feature maps. Consequently, even we remove the corresponding filter, the represented information and knowledge of its generated low-independence feature map can still be largely preserved and approximately reconstructed by other feature maps of other filters after the fine-tuning procedure. In other words, the filters that generate low-independence feature maps tend to exhibit more "replaceability", which can be interpreted as lower importance. Therefore, removing those filters with low channel-independence feature maps will be safe while still preserving high model capacity.

**How to Measure Channel Independence?** Next we discuss how to properly measure the independence of one feature map from others. To that end, four important questions need to be answered.

**Question #1:** *Which mathematical metric should be adopted to quantify the independence of one feature map from other feature maps?*

Analysis. Considering the entire set of feature maps generated from one layer is a 3-D tensor, we propose to extract the linear dependence information of each feature map within the framework of linear algebra. To be specific, given output feature map set of the $l$-th layer $\mathcal{A}^l$, we first matricize $\mathcal{A}^l$ to $A^l = [a_1^{l\ T}, a_2^{l\ T}, \cdots, a_{c^l}^{l\ T}]^T \in \mathbb{R}^{c^l \times hw}$, where a row vector $a_i^l \in \mathbb{R}^{hw}$ is the vectorized $A_i^l$. In such a scenario, the linear independence of each vectorized feature map $a_i^l$, as a row of the matricized entire set of feature maps $A^l$, can be measured via the existing matrix analysis tool.

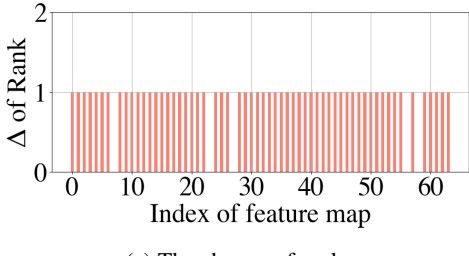

(a) The change of rank.

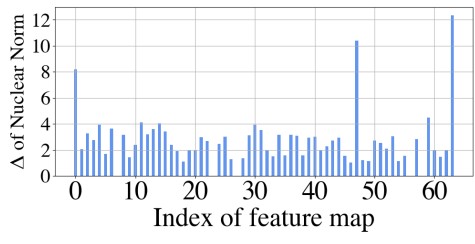

(b) The change of nuclear norm.

Figure 2: The **change** of (a) rank and (b) nuclear norm of the entire set of feature maps ($\boldsymbol{A}^l$) when one feature map ($\boldsymbol{a}_i^l$) is removed. The x-axis represents the index of the feature map that is removed. The y-axis represents the corresponding rank/nuclear norm change of the entire set of feature maps. The feature maps are output from one layer of the ResNet-50 model with input as ImageNet image. **It is seen that change of nuclear norm can better reveal the impact of the deleted feature map on the entire set of feature maps.**

The most straightforward solution is to use *rank* to determine the independence of $\boldsymbol{a}_i^l$, since rank mathematically represents the maximum number of linearly independent rows/columns of the matrix. For instance, we can remove one row from the matrix, and calculate the rank change of the matrix, and then identify the impact and the importance of the deleted row – the less rank change, the less independence (and the importance) of the removed row.

Our Proposal. However, in the context of filter pruning, we believe, the ***change of nuclear norm of the entire set of feature maps***, is a better metric to quantify the independence of each feature map. This is because, as the $\ell_1$-norm of singular values of the matrix, the nuclear norm can reveal richer "soft" information on the impact of the deleted row on the matrix; while the rank, as the $\ell_0$-norm of the singular values, is too "hard" to reflect such change. For instance, as shown in Fig. 2, when we select $\boldsymbol{a}_i^l$, as one row of $\boldsymbol{A}^l$, to be removed, the rank change of $\boldsymbol{A}^l$ is almost the same regardless of our selection of $\boldsymbol{a}_i^l$; while the corresponding changes of nuclear norm vary significantly when different $\boldsymbol{a}_i^l$ are deleted. Therefore, the change of nuclear norm can be viewed as a more precise metric to measure the linear independence of one feature map in a more fine-grained way. In general, the channel independence of one feature map is defined and calculated as below:

**Definition 1 (Channel independence of single feature map)** *For the $i$-th layer with output feature maps $\boldsymbol{\mathcal{A}}^l = \{\boldsymbol{A}_1^l, \boldsymbol{A}_2^l, \cdots, \boldsymbol{A}_{c^l}^l\} \in \mathbb{R}^{c^l \times h \times w}$, the Channel Independence (CI) of one feature map $\boldsymbol{A}_i^l \in \mathbb{R}^{h \times w}$ in the $i$-th channel is defined and calculated as:*

$$CI(\boldsymbol{A}_i^l) \triangleq \|\boldsymbol{A}^l\|_* - \|\boldsymbol{M}_i^l \odot \boldsymbol{A}^l\|_*, \tag{3}$$

where $\boldsymbol{A}^l \in \mathbb{R}^{c^l \times hw}$ is the matricized $\boldsymbol{\mathcal{A}}^l$, $\|\cdot\|_*$ is the nuclear norm, $\odot$ is the Hadamard product, and $\boldsymbol{M}_i^l \in \mathbb{R}^{c^l \times hw}$ is the *row mask matrix* whose $i$-th row entries are zeros and other entries are ones.

**Question #2:** *What is the proper scheme to quantify the independence of multiple feature maps?*

Analysis. Eq. 3 describes the measurement of channel independence for a single feature map. However, in practice filter pruning typically aims to remove multiple filters, which means the independence of the combination of multiple feature maps needs to be calculated. In the case of pruning $m$ filters, such scenario corresponds to checking the changes of nuclear norm of the original $c^l$-row $\boldsymbol{A}^l$ after removing $m$ rows ($\boldsymbol{a}_i^l$). A straightforward solution is to just calculate $C_m^{c^l}$ changes of nuclear norms for all the possible $m$-row removal choices, and then select the one which corresponds to the smallest change. However, this strategy is very computationally expensive, and sometimes even intractable when $c^l$ is large. For instance, in order to identify the smallest nuclear norm change for pruning 50% filters of a 256-output channel ResNet-50 layer, such brutal-force measurement requires more than $5 \times 10^{75}$ times of nuclear norm calculation.

Our Proposal. To address this computational challenge, we propose to leverage the independence of individual feature map to *approximate* the independence of their combination. To be specific, in order to determine $m$ least independent rows in the $\boldsymbol{A}^l$, we first iteratively remove one row ($\boldsymbol{a}_i^l$) from $\boldsymbol{A}^l$ and calculate the corresponding nuclear norm change between the remaining ($c^l - 1$)-row matrix

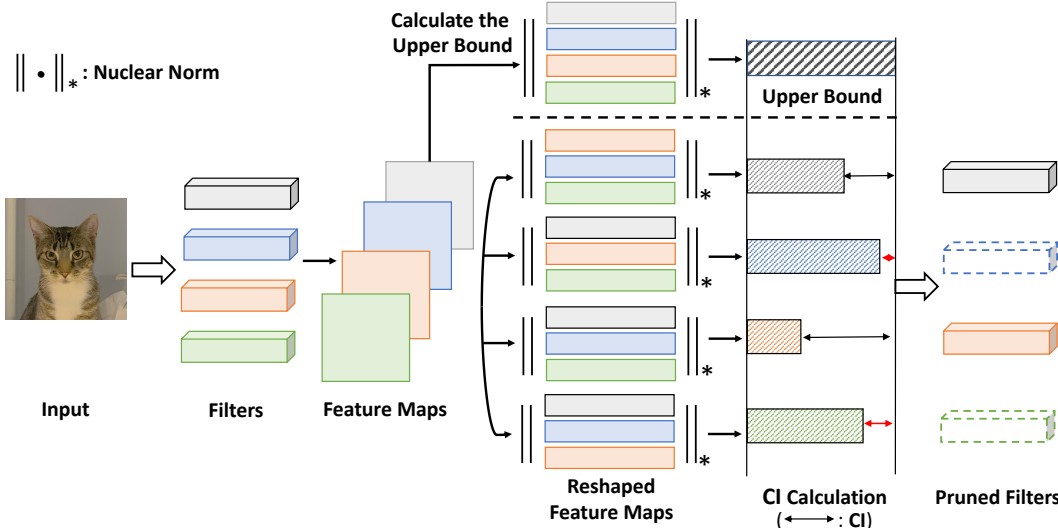

Figure 3: Example of filter pruning process using the change of nuclear norm-based channel independence (CI) criterion.

and the original $c^l$-row $\boldsymbol{A}^l$. Then, among the $c^l$ calculated changes, we identify the $m$ smallest ones and the corresponding removed $\boldsymbol{a}_i^l$. Those selected $m$ vectorized feature maps $\boldsymbol{a}_i^l$ are interpreted as the less independent from other feature maps, and hence their corresponding filters $\mathcal{F}_i^l$ are less important ones that should be pruned. In general, this individual independence-based measurement can closely approximate the combined independence of multiple feature maps (see **Definition 2**). Such approximation requires much less computational complexity (reduction from $\mathcal{O}(C(N, \kappa))$ to $\mathcal{O}(N)$); while still achieving superior filter pruning performance (see Section 4 for evaluation results).

**Definition 2 (Approximated channel independence of combined multiple feature maps)** *For the $i$-th layer with output feature maps $\boldsymbol{A}^l \in \mathbb{R}^{c^l \times h \times w}$, the channel independence of combined $m$ feature maps $\{\boldsymbol{A}_{b_i}^l\}_{i=1}^m$, where $\boldsymbol{A}_{b_i}^l \in \mathbb{R}^{h \times w}$ is in the $b_i$-th channel, is defined and approximated as:*

$$CI(\{\boldsymbol{A}_{b_i}^l\}_{i=1}^m) \triangleq \|\boldsymbol{A}^l\|_* - \|\boldsymbol{M}_{b_1,\cdots,b_m}^l \odot \boldsymbol{A}^l\|_* \approx \sum_{i=1}^m CI(\boldsymbol{A}_{b_i}^l), \qquad (4)$$

where $\boldsymbol{M}_{b_1,\cdots,b_m}^l$ is the multi-row mask matrix, in which the $b_1, \cdots, b_m$-th row entries are zeros and all the other entries are ones.

**Question #3:** *How is the sensitiveness of channel independence related to the distribution of input data?*

Our Observation. Consider our proposed channel independence-based filter pruning is a data-driven approach, its reliability with different distributions of input data should be carefully ensured and examined. To that end, we perform empirical evaluations on channel independence with respect to multiple input images. We observe that the *average* channel independence of each feature map is very stable *at the batch level*. In other words, we can simply input small batches of image samples, and calculate the average channel independence, and then such averaged channel independence with a small number of input data can be used to estimate the channel independence with all the input data. As illustrated in Fig. 4, for the same feature map, the average channel independence in different batches remains very similar, thereby indicating that our channel independence-based approach is robust against different input data.

**Question #4:** *Is this one-shot importance determination scheme good enough? Do we need to further learn and adjust the pruning mask from the data?*

Our Observation. As described above, our proposed scheme calculates the channel independence to identify the filter importance. Considering our approach is built on one-shot calculation, a natural extension is to further adjust the importance ranking via additional learning. To be specific, if we interpret the filter pruning is a *channel-wise masking operation* over the entire weight tensor, the

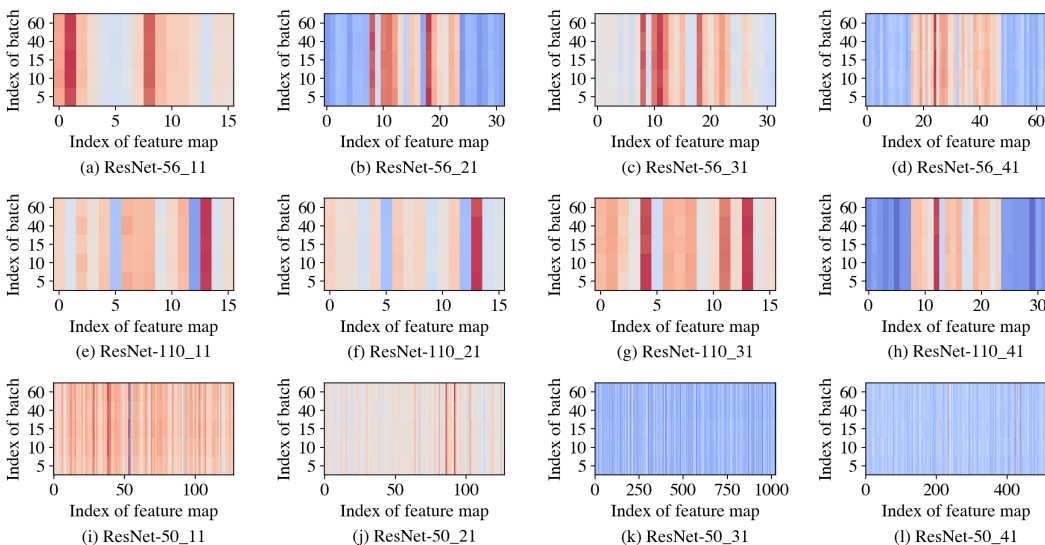

Figure 4: The channel independence of feature maps for one layer in ResNet-50. Here the channel independence is averaged for one batch of input images. The x-axis is the index of the feature map. The y-axis is the index of batches of input images. Here the batch size is 128. **Different colors denote the different values of channel independence. It is seen that the average channel independence is very stable regardless of different input data batches.**

---

**Algorithm 1** CHannel Independence-based Pruning (CHIP) procedure for the $l$-th layer

---

**Input:** Pre-trained weight tensor $\mathcal{W}^l$, $N$ sets of feature maps $\mathcal{A}^l = \{\boldsymbol{A}_1^l, \boldsymbol{A}_2^l, \cdots, \boldsymbol{A}_{c^l}^l\} \in \mathbb{R}^{c^l \times h \times w}$
    from $N$ input samples, and the desired number of filters to be preserved $\kappa^l$.
**Output:** Pruned weight tensor $\mathcal{W}_{prune}^l$.
 1: **for** each input sample **do**
 2:    **Flatten feature maps:** $\boldsymbol{A}^l := \texttt{reshape}(\mathcal{A}^l, [c^l, hw])$;
 3:    **for** $i = 1$ to $c^l$ **do**
 4:        **CI calculation:** Calculate $CI(\boldsymbol{A}_i^l)$ via Equation 3;
 5:    **end for**
 6: **end for**
 7: **Averaging:** Average $CI(\boldsymbol{A}_i^l)$ under all $N$ input samples;
 8: **Sorting:** Sort $\{CI(\boldsymbol{A}_i^l)\}_{i=1}^{c^l}$ in ascending order;
 9: **Pruning**: Prune $c^l - \kappa^l$ filters in $\mathcal{W}^l$ corresponding to the $c^l - \kappa^l$ smallest $CI(\boldsymbol{A}_i^l)$;
10: **Fine-tuning:** Obtain final $\mathcal{W}_{prune}^l$ via fine-tuning $\mathcal{W}^l$ with removing the pruned filter channels.

---

selection of channel mask can be learned from data, and such learning process can use the pruning mask determined by our approach as the initialization. Though in principle this learning-based strategy is expected to enable additional performance improvement, our empirical evaluations show that the consecutive learning procedure does not easily bring further accuracy increase (with the target compression ratio) or compression ratio increase (with the target accuracy) – more experimental details are reported in Supplementary Material. We hypothesize the reason for such phenomenon is that, our proposed nuclear norm change-based channel independence, though only requires one-time calculation, already identifies and captures the importance of feature maps (and its corresponding filters) with high quality, and hence further learning-based adjustment of pruning mask does not easily provide additional improvement.

**The Overall Algorithm.** After addressing the above four problems, we can then integrate our proposals and observations to develop the entire filter pruning procedure from the inter-channel perspective. Algorithm 1 describes and summarizes the overall scheme for our proposed CHannel Independence-based filter Pruning (CHIP) algorithm.

# 4 Experiments

## 4.1 Experimental Settings

**Baselines Models and Datasets.** To demonstrate the effectiveness and generality of our proposed channel independence-based approach, we evaluate its pruning performance for various baseline models on different image classification datasets. To be specific, we conduct experiments for three CNN models (ResNet-56, ResNet-110 and VGG-16) on CIFAR-10 dataset [24]. Also, we further evaluate our approach and compare its performance with other state-of-the-art pruning methods for ResNet-50 model on large-scale ImageNet dataset [5].

**Pruning and Fine-tuning Configurations.** We conduct our empirical evaluations on Nvidia Tesla V100 GPUs with PyTorch 1.7 framework. To determine the importance of each filter, we randomly sample 5 batches (640 input images) to calculate the average channel independence of each feature map in all the experiments. After performing the channel independence-based filter pruning, we then perform fine-tuning on the pruned models with Stochastic Gradient Descent (SGD) as the optimizer. To be specific, we perform the fine-tuning for 300 epochs on CIFAR-10 datasets with the batch size, momentum, weight decay and initial learning rate as 128, 0.9, 0.05 and 0.01, respectively. On the ImageNet dataset, fine-tuning is performed for 180 epochs with the batch size, momentum, weight decay and initial learning rate as 256, 0.99, 0.0001 and 0.1, respectively.

Table 1: Experimental results on CIFAR-10 dataset.

| Method | Top-1 Accuracy (%) | | | # Params. ($\downarrow$%) | FLOPs ($\downarrow$%) |
| --- | --- | --- | --- | --- | --- |
| | Baseline | Pruned | $\Delta$ | | |
| ResNet-56 | | | | | |
| $\ell_1$-norm (2016) [27] | 93.04 | 93.06 | +0.02 | 0.73M(13.7) | 90.90M(27.6) |
| NISP (2018) [59] | 93.04 | 93.01 | -0.03 | 0.49M(42.4) | 81.00M(35.5) |
| GAL (2019) [32] | 93.26 | 93.38 | +0.12 | 0.75M(11.8) | 78.30M(37.6) |
| HRank (2020) [31] | 93.26 | 93.52 | +0.26 | 0.71M(16.8) | 88.72M(29.3) |
| **CHIP (Ours)** | 93.26 | **94.16** | **+0.90** | **0.48M(42.8)** | **65.94M(47.4)** |
| GAL (2019) [32] | 93.26 | 91.58 | -1.68 | 0.29M(65.9) | 49.99M(60.2) |
| LASSO (2017) [21] | 92.80 | 91.80 | -1.00 | N/A | 62.00M(50.6) |
| HRank (2020) [31] | 93.26 | 90.72 | -2.54 | 0.27M(68.1) | 32.52M(74.1) |
| **CHIP (Ours)** | 93.26 | **92.05** | -1.21 | **0.24M(71.8)** | 34.79M(72.3) |
| ResNet-110 | | | | | |
| $\ell_1$-norm (2016) [27] | 93.53 | 93.30 | -0.23 | 1.16M(32.4) | 155.00M(38.7) |
| HRank (2020) [31] | 93.50 | 94.23 | +0.73 | 1.04M(39.4) | 148.70M(41.2) |
| **CHIP (Ours)** | 93.50 | **94.44** | **+0.94** | **0.89M(48.3)** | **121.09M(52.1)** |
| GAL (2019) [32] | 93.50 | 92.74 | -0.76 | 0.95M(44.8) | 130.20M(48.5) |
| HRank (2020) [31] | 93.50 | 92.65 | -0.85 | 0.53M(68.7) | 79.30M(68.6) |
| **CHIP (Ours)** | 93.50 | **93.63** | **+0.13** | 0.54M(68.3) | **71.69M(71.6)** |
| VGG-16 | | | | | |
| SSS (2018) [23] | 93.96 | 93.02 | -0.94 | 3.93M(73.8) | 183.13M(41.6) |
| GAL (2019) [32] | 93.96 | 93.77 | -0.19 | 3.36M(77.6) | 189.49M(39.6) |
| HRank (2020) [31] | 93.96 | 93.43 | -0.53 | 2.51M(82.9) | 145.61M(53.5) |
| **CHIP (Ours)** | 93.96 | **93.86** | **-0.10** | 2.76M(81.6) | **131.17M(58.1)** |
| GAL (2019) [32] | 93.96 | 93.42 | -0.54 | 2.67M(82.2) | 171.89M(45.2) |
| HRank (2020) [31] | 93.96 | 92.34 | -1.62 | 2.64M(82.1) | 108.61M(65.3) |
| **CHIP (Ours)** | 93.96 | **93.72** | **-0.24** | **2.50M(83.3)** | **104.78M(66.6)** |
| HRank (2020) [31] | 93.96 | 91.23 | -2.73 | 1.78M(92.0) | 73.70M(76.5) |
| **CHIP (Ours)** | 93.96 | **93.18** | **-0.78** | 1.90M(87.3) | **66.95M(78.6)** |

## 4.2 Evaluation and Comparison on CIFAR-10 Dataset

Table 1 shows the evaluation results of the pruned ResNet-56, ResNet-110 and VGG-16 models on CIFAR-10 dataset. To be consistent with prior works, we evaluate the performance for two scenarios: targeting high accuracy and targeting high model size and FLOPs reductions.

**ResNet-56.** For ResNet-56 model, our channel independence-based approach can bring 0.90% accuracy increase over the baseline model with 42.8% and 47.4% model size and FLOPs reductions, respectively. When we adopt aggressive compression with 71.8% and 72.3% model size and FLOPs reductions, we can still achieve high performance – our solution enables 1.33% higher accuracy than HRank [31] with the similar model size and computational costs.

**ResNet-110.** For ResNet-110 model, our approach can bring 0.94% accuracy increase over the baseline model with 48.3% and 52.1% model size and FLOPs reductions, respectively. When we perform aggressive pruning with 68.3% and 71.6% model size and FLOPs reductions, our pruned model can still achieve 0.13% higher accuracy over the baseline model.

**VGG-16.** For VGG-16 model, our approach can bring 81.6% and 58.1% model size and FLOPs reductions, respectively, with only 0.1% accuracy drop. Moreover, with 83.3% and 66.6% storage and computational cost reductions, our pruned model can achieve 1.38% higher accuracy than the model using other pruning approaches under a similar compression ratio. For even higher FLOPs reduction (78.6%), our method can bring nearly 2% accuracy increase over the prior works.

## 4.3 Evaluation and Comparison on ImageNet Dataset

Table 2 summarizes the pruning performance of our approach for ResNet-50 on ImageNet dataset. It is seen that when targeting a moderate compression ratio, our approach can achieve 40.8% and 44.8% storage and computation reductions, respectively, with 0.15% accuracy increase over the baseline model. When we further increase the compression ratio, our approach still achieves superior performance than state-of-the-art works. For instance, compared with SCOP [51], our approach shows higher accuracy (0.12%) in moderate compression region and the same accuracy in high compress region; while meanwhile enjoying a much smaller model size and fewer FLOPs.

Table 2: Experimental results on ImageNet dataset.

| Method | Top-1 Accuracy (%) | | | Top-5 Accuracy (%) | | | Params. ↓(%) | FLOPs ↓(%) |
|---|---|---|---|---|---|---|---|---|
| | Baseline | Pruned | Δ | Baseline | Pruned | Δ | | |
| ResNet-50 | | | | | | | | |
| ThiNet (2017) [37] | 72.88 | 72.04 | -0.84 | 91.14 | 90.67 | -0.47 | 33.7 | 36.8 |
| SFP (2018)[19] | 76.15 | 74.61 | -1.54 | 92.87 | 92.06 | -0.81 | N/A | 41.8 |
| Autopruner (2020) [36] | 76.15 | 74.76 | -1.39 | 92.87 | 92.15 | -0.72 | N/A | 48.7 |
| FPGM (2019) [20] | 76.15 | 75.59 | -0.56 | 92.87 | 92.63 | -0.24 | 37.5 | 42.2 |
| Taylor (2019) [38] | 76.18 | 74.50 | -1.68 | N/A | N/A | N/A | 44.5 | 44.9 |
| C-SGD (2019) [7] | 75.33 | 74.93 | -0.40 | 92.56 | 92.27 | -0.29 | N/A | 46.2 |
| GAL (2019) [32] | 76.15 | 71.95 | -4.20 | 92.87 | 90.94 | -1.93 | 16.9 | 43 |
| RRBP (2019) [61] | 76.10 | 73.00 | -3.10 | 92.90 | 91.00 | -1.90 | N/A | 54.5 |
| PFP (2020) [30] | 76.13 | 75.91 | -0.22 | 92.87 | 92.81 | -0.06 | 18.1 | 10.8 |
| HRank (2020) [31] | 76.15 | 74.98 | -1.17 | 92.87 | 92.33 | -0.54 | 36.6 | 43.7 |
| SCOP (2020) [51] | 76.15 | 75.95 | -0.20 | 92.87 | 92.79 | -0.08 | 42.8 | 45.3 |
| **CHIP (Ours)** | 76.15 | **76.30** | **+0.15** | 92.87 | **93.02** | **+0.15** | 40.8 | 44.8 |
| **CHIP (Ours)** | 76.15 | 76.15 | 0.00 | 92.87 | 92.91 | +0.04 | **44.2** | **48.7** |
| PFP (2020) [30] | 76.13 | 75.21 | -0.92 | 92.87 | 92.43 | -0.44 | 30.1 | 44 |
| SCOP (2020) [51] | 76.15 | 75.26 | -0.89 | 92.87 | 92.53 | -0.34 | 51.8 | 54.6 |
| **CHIP (Ours)** | 76.15 | **75.26** | **-0.89** | 92.87 | **92.53** | **-0.34** | **56.7** | **62.8** |
| HRank (2020) [31] | 76.15 | 71.98 | -4.17 | 92.87 | 91.01 | -1.86 | 46.0 | 62.1 |
| HRank (2020) [31] | 76.15 | 69.10 | -7.05 | 92.87 | 89.58 | -3.29 | 67.5 | 76.0 |
| **CHIP (Ours)** | 76.15 | **73.30** | **-2.85** | 92.87 | **91.48** | **-1.39** | **68.6** | **76.7** |

# 5 Conclusion

In this paper, we propose to use channel independence, an inter-channel perspective-motivated metric, to evaluate the importance of filters for network pruning. By systematically exploring the quantification metric, measuring scheme, and sensitiveness and reliability of channel independence, we develop CHIP, a CHannel Independence-based filter pruning for neural network compression. Extensive evaluation results on different datasets show our proposed approach brings significant storage and computational cost reductions while still preserving high model accuracy.

## Broader Impact

As technology advances, cell phones, laptops, wearable gadgets and intelligent connected vehicles with specific chips are required to handle more complicated tasks by deploying neural networks. However, more powerful networks will cost more memory size and running time. Network pruning is the main strategy to reduce the memory size and accelerate the run-time during the inference stage. Benefiting from pruning techniques and specific designs for hardware [62, 4], IoT (Internet of Things) devices are able to execute complex projects based on small and efficient models.

## Acknowledgements and Funding Disclosure

Bo Yuan would like to thank the support from National Science Foundation (NSF) award CCF-1937403. Saman Zonouz would like to thank the support from NSF CPS and SATC programs.

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
