# Supplementary Material

# 1 Additional Studies

Besides the evaluation of various models on different datasets, we also perform additional studies to obtain deep understandings of our proposed channel independence-based filter pruning approach.

## 1.1 Relationship between Channel Independence and Importance of Feature Map

We use a numerical example to demonstrate the relationship between Channel Independence (CI) and importance of feature maps. Here for the following example $3\times4$ matrix, each of its rows denotes one vectorized feature map of one channel. Our goal is to identify the least important row that can be represented by other rows. Intuitively, Row-1 or Row-2 should be removed due to their linear dependence. Furthermore, because the $l_2$-norm of Row-2 is less than that of Row-1, Row-2 is expected to be the least important one.

$$\begin{pmatrix} 0.9 & 0.8 & 1.1 & 1.2 \\ 0.81 & 0.72 & 0.99 & 1.08 \\ 0.8 & 0.9 & 1.2 & 1.1 \end{pmatrix} \tag{1}$$

Now according to Equation 3, we can obtain the CI of each row as shown in Table 1:

Table 1: CI of each row.

| | |
|---|---|
| CI of Row-1 | 0.696 |
| CI of Row-2 | 0.549 |
| CI of Row-3 | 0.827 |

And it is seen that Row-2 is assigned as the smallest CI, which is consistent with our expectation.

## 1.2 Balance between Pruning and Task Performance

In the context of model compression, high pruning rate and high accuracy cannot be always achieved at the same time – an efficient compression approach should provide good balance between compression performance and task performance. Fig. 1 shows the change of accuracy of the pruned ResNet-50 on ImageNet dataset via using our approach with respect to different pruning ratios. It can be seen that our approach can effectively reduce the number of model parameters and FLOPs with good performance on test accuracy.

## 1.3 Accuracy-Pruning Rate Trade-off Curves of Different Pruning Methods

We study the accuracy-pruning rate trade-off curves of different pruning methods (CHIP, SCOP, HRank) for ResNet-50 on ImageNet. The results are shown in Fig. 2.

## 1.4 Quantified Sensitiveness of Channel Independence to Input Data

To analyze the potential sensitiveness of channel independence to input data (as indicated in **Question #3**), Fig. 4 in the main paper visualizes the average channel independence with different batches of input images to show that the channel independence is not sensitive to the change of inputs. In this supplementary material, we further quantify the sensitiveness. To be specific, for each batch of input data (batch size = 128), we form a length-64 vector consisting of the average channel independence for all the 64 feature maps of one layer (ResNet-56_55) in ResNet-56 model on CIFAR-10 dataset, and then we calculate the **Pearson correlation coefficient** among different channel independence vectors that correspond to different batches. As shown in Table 2, those vectors are highly correlated with each other though they are generated from different input batches, thereby demonstrating the low sensitiveness of channel independence metric to the input data.

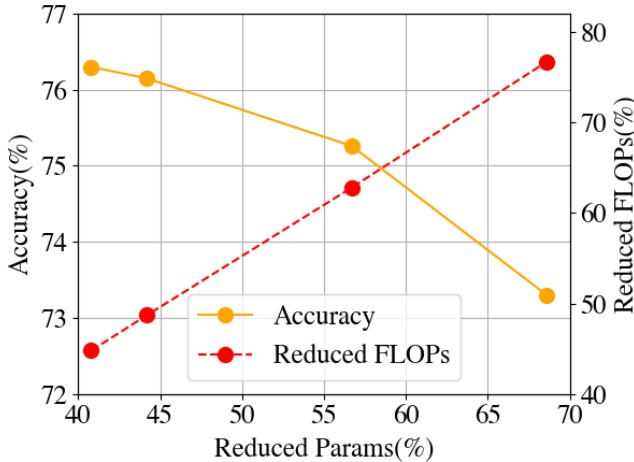

Figure 1: The accuracies and computational costs of our pruned ResNet-50 model with respect to different pruning ratios (on ImageNet dataset).

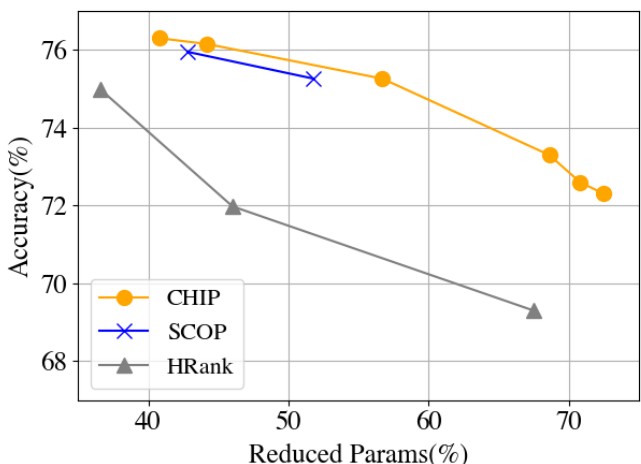

Figure 2: The accuracies of ResNet-50 model from different methods (CHIP, SCOP, HRank) with respect to different pruning ratios (on ImageNet dataset).

## 1.5   Is Additional Adjustment of Importance Ranking Needed?

As analyzed in **Question #4**, a potential extension of our approach is to introduce an additional phase to further adjust the importance ranking from the training data, once our one-shot channel independence-based pruning is finished. To be specific, an even better channel-wise pruning mask strategy could be further learned built upon the mask determined by our approach as the initialization. Intuitively, this data-driven strategy might potentially provide an extra performance improvement.

To explore this potential opportunity, we conduct experiments for different models on different datasets. Our empirical observation is that an additional learning phase for the pruning mask does not bring an extra accuracy increase. Fig. 3 visualizes the same part of filters in Conv1 layer of VGG-16 without and with additional pruning mask training. It is seen that there is nothing change for the selected filters to be pruned before and after using the trained mask. Our experiments for other models on other datasets also show the same phenomenon. Therefore we conclude that additional adjustment on the pruning mask is not required in the context of our channel independence-based filter pruning.

Table 2: Pearson correlation coefficient among 5 length-64 different channel independence vectors of ResNet-56_55 layer (containing 64 output feature maps) with 5 different input batches (CIFAR-10 dataset).

| - | Vector-1 | Vector-2 | Vector-3 | Vector-4 | Vector-5 |
|---|---|---|---|---|---|
| Vector-1 | 1 | 0.907 | 0.850 | 0.911 | 0.821 |
| Vector-2 | 0.907 | 1 | 0.880 | 0.899 | 0.901 |
| Vector-3 | 0.850 | 0.880 | 1 | 0.913 | 0.913 |
| Vector-4 | 0.911 | 0.899 | 0.913 | 1 | 0.881 |
| Vector-5 | 0.821 | 0.901 | 0.913 | 0.881 | 1 |

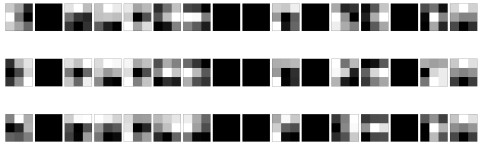 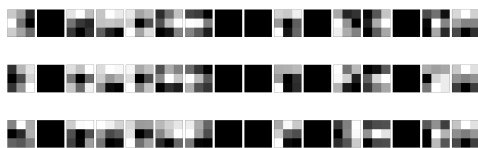

(a) Visualization of filters without further pruning mask adjustment.

(b) Visualization of filters with further pruning mask adjustment.

Figure 3: Visualization of filters in Conv1 layer of VGG-16 model on CIFAR-10 dataset. Here we only show the first 16 out of 64 filters of this layer due to the space limitation. **Left:** the pruned filters using our approach. **Right:** the pruned filters after further pruning mask adjustment with the mask determined by our approach as initialization. x-axis represents different filters and y-axis represents different input channels. The kernel size is $3 \times 3$. Black kernels are the pruned ones.

How to find the best combination of the largest $CI(\{\boldsymbol{A}_{b_i}^l\}_{i=1}^m)$? Given one image randomly sampled from total images, metricized feature maps $\{\boldsymbol{A}_{b_i}^l\}_{i=1}^m$ of $l$-th layer are generated after the inference. Firstly, we calculate the $CI$ upon our Algorithm 1. Then, we initial the score of $\boldsymbol{M}_{b_1,\cdots,b_m}^l$ based on the normalized $CI$. That is, if we are not going to train the $\boldsymbol{M}_{b_1,\cdots,b_m}^l$ to change $b_1,\cdots,b_m$, the nuclear norm and index of pruned filters from $\{\boldsymbol{A}_{b_i}^l\}_{i=1}^m$ equals to the result from our Algorithm 1. Therefore, this initialization can be viewed as a baseline for $CI(\boldsymbol{A}_{b_i}^l)$. Secondly, we train the $\boldsymbol{M}_{b_1,\cdots,b_m}^l$ using the MSE loss functions to minimize the gap between the Upper Bound and current nuclear norm under sparsity 83.3% from VGG-16. With optimizer of ADAM and SGD, the learning rate is set from 0.1 to 0.001 and the weight decay is set from 0.05 to 5. Among each possible pair of above hyperparameters, we get the pruned filters of maximal $CI(\{\boldsymbol{A}_{b_i}^l\}_{i=1}^m)$ from what we desire.

To sum up, although there has not been proven theoretically, we find that index of pruned filter generated from our method almost equals to index of pruned filters from global optimal methods in experiments.

## 2 Detailed Setting of $\kappa^l$ and Pruning Ratios

In this section, we provide the details of $\kappa^l$ (number of preserved filters) and pruning ratios of all layers. On CIFAR-10, we report the $\kappa^l$ and pruning ratios for ResNet-56, ResNet-110 and VGG-16. On ImageNet, $\kappa^l$ and pruning ratios are reported for ResNet-50.

### 2.1 $\kappa^l$ (Number of Preserved Filters of All Layers)

#### 2.1.1 ResNet-56

**For overall sparsity 42.8%, layer-wise $\kappa^l$ are :** [16, 9, 13, 9, 13, 9, 13, 9, 13, 9, 13, 9, 13, 9, 13, 9, 13, 9, 13, 19, 27, 19, 27, 19, 27, 19, 27, 19, 27, 19, 27, 19, 27, 19, 27, 38, 64, 38, 64, 38, 64, 38, 64, 38, 64, 38, 64, 38, 64, 38, 64, 38, 64]

**For overall sparsity 71.8%, layer-wise** $\kappa^l$ **are :** [16, 8, 9, 8, 9, 8, 9, 8, 9, 8, 9, 8, 9, 8, 9, 8, 9, 8, 9, 12, 19, 12, 19, 12, 19, 12, 19, 12, 19, 12, 19, 12, 19, 12, 19, 12, 19, 19, 64, 19, 64, 19, 64, 19, 64, 19, 64, 19, 64, 19, 64, 19, 64, 19, 64]

### 2.1.2 ResNet-110

**For overall sparsity 48.3%, layer-wise** $\kappa^l$ **are :** [16, 10, 12, 10, 12, 10, 12, 10, 12, 10, 12, 10, 12, 10, 12, 10, 12, 10, 12, 10, 12, 10, 12, 10, 12, 10, 12, 10, 12, 10, 12, 10, 12, 10, 12, 10, 12, 17, 24, 17, 24, 17, 24, 17, 24, 17, 24, 17, 24, 17, 24, 17, 24, 17, 24, 17, 24, 17, 24, 17, 24, 17, 24, 17, 24, 35, 64, 35, 64, 35, 64, 35, 64, 35, 64, 35, 64, 35, 64, 35, 64, 35, 64, 35, 64, 35, 64, 35, 64, 35, 64, 35, 64, 35, 64, 35, 64]

**For overall sparsity 68.3%, layer-wise** $\kappa^l$ **are :** [16, 8, 9, 8, 9, 8, 9, 8, 9, 8, 9, 8, 9, 8, 9, 8, 9, 8, 9, 8, 9, 8, 9, 8, 9, 8, 9, 8, 9, 8, 9, 8, 9, 8, 9, 8, 9, 11, 19, 11, 19, 11, 19, 11, 19, 11, 19, 11, 19, 11, 19, 11, 19, 11, 19, 11, 19, 11, 19, 11, 19, 11, 19, 11, 19, 11, 19, 11, 19, 22, 64, 22, 64, 22, 64, 22, 64, 22, 64, 22, 64, 22, 64, 22, 64, 22, 64, 22, 64, 22, 64, 22, 64, 22, 64, 22, 64, 22, 64]

### 2.1.3 VGG-16

**For overall sparsity 81.6%, layer-wise** $\kappa^l$ **are :** [50, 50, 101, 101, 202, 202, 202, 128, 128, 128, 128, 128, 512]

**For overall sparsity 83.3%, layer-wise** $\kappa^l$ **are :** [44, 44, 89, 89, 179, 179, 179, 128, 128, 128, 128, 128, 512]

**For overall sparsity 87.3%, layer-wise** $\kappa^l$ **are :** [35, 35, 70, 70, 140, 140, 140, 112, 112, 112, 112, 112, 512]

### 2.1.4 ResNet-50

**For overall sparsity 40.8%, layer-wise** $\kappa^l$ **are :** [64, 41, 41, 230, 41, 41, 230, 41, 41, 230, 83, 83, 460, 83, 83, 460, 83, 83, 460, 83, 83, 460, 166, 166, 912, 166, 166, 912, 166, 166, 912, 166, 166, 912, 166, 166, 912, 166, 166, 912, 332, 332, 2048, 332, 332, 2048, 332, 332, 2048, 332, 332, 2048]

**For overall sparsity 44.2%, layer-wise** $\kappa^l$ **are :** [64, 39, 39, 225, 39, 39, 225, 39, 39, 225, 79, 79, 450, 79, 79, 450, 79, 79, 450, 79, 79, 450, 158, 158, 901, 158, 158, 901, 158, 158, 901, 158, 158, 901, 158, 158, 901, 158, 158, 901, 317, 317, 2048, 317, 317, 2048, 317, 317, 2048]

**For overall sparsity 56.7%, layer-wise** $\kappa^l$ **are :** [64, 32, 32, 192, 32, 32, 192, 32, 32, 192, 64, 64, 384, 64, 64, 384, 64, 64, 384, 64, 64, 384, 128, 128, 768, 128, 128, 768, 128, 128, 768, 128, 128, 768, 128, 128, 768, 128, 128, 768, 256, 256, 2048, 256, 256, 2048, 256, 256, 2048]

**For overall sparsity 68.6%, layer-wise** $\kappa^l$ **are :** [64, 25, 25, 128, 25, 25, 128, 25, 25, 128, 51, 51, 256, 51, 51, 256, 51, 51, 256, 51, 51, 256, 102, 102, 512, 102, 102, 512, 102, 102, 512, 102, 102, 512, 102, 102, 512, 102, 102, 512, 204, 204, 2048, 204, 204, 2048, 204, 204, 2048]

## 2.2 Pruning Ratios (Sparsity) of All Layers

### 2.2.1 ResNet-56

**For overall sparsity 42.8%, layer-wise pruning ratios are :** [0.0, 0.4, 0.15, 0.4, 0.15, 0.4, 0.15, 0.4, 0.15, 0.4, 0.15, 0.4, 0.15, 0.4, 0.15, 0.4, 0.15, 0.4, 0.15, 0.4, 0.15, 0.4, 0.15, 0.4, 0.15, 0.4, 0.15, 0.4, 0.15, 0.4, 0.15, 0.4, 0.15, 0.4, 0.15, 0.4, 0.15, 0.4, 0.0, 0.4, 0.0, 0.4, 0.0, 0.4, 0.0, 0.4, 0.0, 0.4, 0.0, 0.4, 0.0, 0.4, 0.0, 0.4, 0.0]

**For overall sparsity 71.8%, layer-wise pruning ratios are :** [0.0, 0.5, 0.4, 0.5, 0.4, 0.5, 0.4, 0.5, 0.4, 0.5, 0.4, 0.5, 0.4, 0.5, 0.4, 0.5, 0.4, 0.5, 0.4, 0.6, 0.4, 0.6, 0.4, 0.6, 0.4, 0.6, 0.4, 0.6, 0.4, 0.6, 0.4, 0.6, 0.4, 0.6, 0.4, 0.6, 0.4, 0.7, 0.0, 0.7, 0.0, 0.7, 0.0, 0.7, 0.0, 0.7, 0.0, 0.7, 0.0, 0.7, 0.0, 0.7, 0.0, 0.7, 0.0]

### 2.2.2 ResNet-110

**For overall sparsity 48.3%, layer-wise pruning ratios are :** [0.0, 0.35, 0.22, 0.35, 0.22, 0.35, 0.22, 0.35, 0.22, 0.35, 0.22, 0.35, 0.22, 0.35, 0.22, 0.35, 0.22, 0.35, 0.22, 0.35, 0.22, 0.35, 0.22, 0.35, 0.22, 0.35, 0.22, 0.35, 0.22, 0.35, 0.22, 0.35, 0.22, 0.35, 0.22, 0.35, 0.22, 0.45, 0.22, 0.45, 0.22, 0.45, 0.22, 0.45, 0.22, 0.45, 0.22, 0.45, 0.22, 0.45, 0.22, 0.45, 0.22, 0.45, 0.22, 0.45, 0.22, 0.45, 0.22, 0.45, 0.22, 0.45, 0.22, 0.45, 0.22, 0.45, 0.22, 0.45, 0.22, 0.45, 0.0, 0.45, 0.0, 0.45, 0.0, 0.45, 0.0, 0.45, 0.0, 0.45, 0.0, 0.45, 0.0, 0.45, 0.0, 0.45, 0.0, 0.45, 0.0, 0.45, 0.0, 0.45, 0.0, 0.45, 0.0, 0.45, 0.0, 0.45, 0.0, 0.45, 0.00]

**For overall sparsity 68.3%, layer-wise pruning ratios are :** [0.0, 0.5, 0.4, 0.5, 0.4, 0.5, 0.4, 0.5, 0.4, 0.5, 0.4, 0.5, 0.4, 0.5, 0.4, 0.5, 0.4, 0.5, 0.4, 0.5, 0.4, 0.5, 0.4, 0.5, 0.4, 0.5, 0.4, 0.5, 0.4, 0.5, 0.4, 0.5, 0.4, 0.5, 0.4, 0.65, 0.4, 0.65, 0.4, 0.65, 0.4, 0.65, 0.4, 0.65, 0.4, 0.65, 0.4, 0.65, 0.4, 0.65, 0.4, 0.65, 0.4, 0.65, 0.4, 0.65, 0.4, 0.65, 0.4, 0.65, 0.4, 0.65, 0.4, 0.65, 0.4, 0.65, 0.4, 0.65, 0.0, 0.65, 0.0, 0.65, 0.0, 0.65, 0.0, 0.65, 0.0, 0.65, 0.0, 0.65, 0.0, 0.65, 0.0, 0.65, 0.0, 0.65, 0.0, 0.65, 0.0, 0.65, 0.0, 0.65, 0.0, 0.65, 0.0, 0.65, 0.0, 0.65, 0.0]

### 2.2.3 VGG-16

**For overall sparsity 81.6%, layer-wise pruning ratios are :** [0.21, 0.21, 0.21, 0.21, 0.21, 0.21, 0.21, 0.75, 0.75, 0.75, 0.75, 0.75, 0]

**For overall sparsity 83.3%, layer-wise pruning ratios are :** [0.3, 0.3, 0.3, 0.3, 0.3, 0.3, 0.3, 0.75, 0.75, 0.75, 0.75, 0.75, 0]

**For overall sparsity 87.3%, layer-wise pruning ratios are :** [0.45, 0.45, 0.45, 0.45, 0.45, 0.45, 0.45, 0.78, 0.78, 0.78, 0.78, 0.78, 0]

### 2.2.4 ResNet-50

**For overall sparsity 40.8%, layer-wise pruning ratios are :** [0.0, 0.35, 0.35, 0.1, 0.35, 0.35, 0.1, 0.35, 0.35, 0.1, 0.35, 0.35, 0.1, 0.35, 0.35, 0.1, 0.35, 0.35, 0.1, 0.35, 0.35, 0.1, 0.35, 0.35, 0.1, 0.35, 0.35, 0.1, 0.35, 0.35, 0.1, 0.35, 0.35, 0.1, 0.35, 0.35, 0.1, 0.35, 0.35, 0.0, 0.35, 0.35, 0.0, 0.35, 0.35, 0.0]

**For overall sparsity 44.2%, layer-wise pruning ratios are :** [0.0, 0.38, 0.38, 0.12, 0.38, 0.38, 0.12, 0.38, 0.38, 0.12, 0.38, 0.38, 0.12, 0.38, 0.38, 0.12, 0.38, 0.38, 0.12, 0.38, 0.38, 0.12, 0.38, 0.38, 0.12, 0.38, 0.38, 0.12, 0.38, 0.38, 0.12, 0.38, 0.38, 0.12, 0.38, 0.38, 0.12, 0.38, 0.38, 0.0, 0.38, 0.38,0.0, 0.38, 0.38, 0.0]

**For overall sparsity 56.7%, layer-wise pruning ratios are :** [0.0, 0.5, 0.5, 0.25, 0.5, 0.5, 0.25, 0.5, 0.5, 0.25, 0.5, 0.5, 0.25, 0.5, 0.5, 0.25, 0.5, 0.5, 0.25, 0.5, 0.5, 0.25, 0.5, 0.5, 0.25, 0.5, 0.5, 0.25, 0.5, 0.5, 0.25, 0.5, 0.5, 0.25, 0.5, 0.5, 0.25, 0.5, 0.5,0.0, 0.5, 0.5,0.0, 0.5, 0.5, 0.0]

**For overall sparsity 68.6%, layer-wise pruning ratios are :** [0.0, 0.6, 0.6, 0.5, 0.6, 0.6, 0.5, 0.6, 0.6, 0.5, 0.6, 0.6, 0.5, 0.6, 0.6, 0.5, 0.6, 0.6, 0.5, 0.6, 0.6, 0.5, 0.6, 0.6, 0.5, 0.6, 0.6, 0.5, 0.6, 0.6, 0.5, 0.6, 0.6, 0.5, 0.6, 0.6, 0.5, 0.6, 0.6, 0.0, 0.6, 0.6, 0.0, 0.6, 0.6, 0.0]