# OpenReview forum: "CHIP: CHannel Independence-based Pruning for Compact Neural Networks"
_NeurIPS.cc/2021/Conference — NeurIPS 2021 Poster_

### Official Review · Reviewer_TZ9P · 2021-07-05

**Rating:** 6
**Confidence:** 4

**Summary:**

The authors propose to explore and leverage the cross-channel feature information for filter pruning. Specifically, the authors propose a cross-channel correlation-based metric to measure the importance of filters. In this case, the low-independence feature maps are considered redundant. As a result, the corresponding filters that output the low-independence feature maps are viewed as uninformative and can be safely removed. Experiments on CIFAR-10 and ImageNet demonstrate the effectiveness of the proposed method. However, the experiments are not sufficient. Please see my detailed comments below.

**Limitations And Societal Impact:**

Please discuss the limitations of the proposed method.

**Main Review:**

**Contributions**:
1.	The authors propose a new metric to measure the correlation of multiple feature maps. Relying on the proposed metric, the proposed method is able to identify important filters from a global perspective.

2.	The authors develop a low-cost fine-grained channel independence metric for efficient filter pruning.

3.	Experiments on CIFAR-10 and ImageNet show that the proposed method is able to achieve high pruning ratios while still achieving promising performance.

**Questions and points needed to be improved**:
1.	CCP [1] also considers the inter-channel relationship during channel pruning. More discussions on the difference between the proposed method and CCP are required. Moreover, it would be better for the authors to provide more discussion on several methods that use gradient-based metrics [2-4] to measure the importance of filters.

2.	In the experiments on ImageNet, it takes too many epochs (180) to finetune the pruned models comparing with other methods (20 in SCOP [5]). It is unclear whether the good performance is resulting from longtime training.

3.	To demonstrate the effectiveness of the proposed pruning metric, it would be better for the authors to show the accuracy-pruning rate trade-off curve of different methods, which will strengthen the paper.

4.	Some important details are missing. How to set the pruning rate of each layer? Do the authors set the same pruning rate for each layer?

5.	In Algorithm 1, the authors compute the channel independence (CI) criterion among $N$ samples. However, the effect of different $N$ is unclear. It would be better for the authors to conduct more experiments to investigate the effect of different $N$.

6.	The authors conduct experiments on heavyweight networks (e.g., ResNet-56, ResNet-110, VGG-16, ResNet-50). However, the performance of the proposed method on lightweight networks is still unknown. It would be better for the authors to conduct more experiments on lightweight models (e.g., MobileNetV1 [6] and MobileNetV2 [7]).

**Reference**:

[1] Collaborative Channel Pruning for Deep Networks. ICML 2019.

[2] Importance Estimation for Neural Network Pruning. CVPR 2019.

[3] SNIP: Single-shot Network Pruning based on Connection Sensitivity. ICLR 2019.

[4] Discrimination-aware Network Pruning for Deep Model Compression. TPAMI 2021.

[5] SCOP: Scientific Control for Reliable Neural Network Pruning. NeurIPS 2020.

[6] MobileNets: Efficient Convolutional Neural Networks for Mobile Vision Applications. arXiv 2017.

[7] MobileNetV2: Inverted Residuals and Linear Bottlenecks. CVPR 2018.


**Time Spent Reviewing:**

20

---

> ### Author Response · Authors · 2021-08-10
> **Response to Reviewer4**
>
> We sincerely appreciate reviewer's very constructive comments and suggestions. The following is our response in order of questions and comments raised.
>
> Q1: More discussions on the difference between the proposed method and CCP are required. Moreover, it would be better for the authors to provide more discussion on several methods that use gradient-based metrics to measure the importance of filters.
>
> Thank you for pointing out these works! We will add a discussion of these papers in the introduction section. Compared with CCP, another approach exploring inter-channel correlation, our approach (CHIP) enjoys two advantages. First, unlike CCP that needs 2nd-order Hessian and 0-1 quadratic minimization, CHIP only needs to simply calculate the nuclear norm of the flatten feature maps. Second, CHIP shows better performance than CCP. For pruning ResNet-50 on ImageNet, with the same baseline 92.87\% top-5 accuracy, CHIP gains 0.04\% accuracy increase with 48.7\% FLOPs reduction; while CCP suffers 0.25\% accuracy drop with 48.8 \% FLOPs reduction. Moreover, compared with other gradient-based approaches that only measure the importance of filters via using intra-channel information, our approach explores utilizing inter-channel correlation for pruning. Our strategy does not need to calculate 1st-order gradient or 2nd-order Hessian but achieves better performance. For instance, [2] (CVPR'19) is included in Table 2 (denoted as Taylor) and its performance is inferior to our approach. Also, compared with [4] (TPAMI'21), to prune ResNet-50 on ImageNet dataset, our approach achieves 62.8\% FLOPs reduction with top-5 accuracy as 92.53\% (our baseline is 92.87\%); while [4] only has 52.41 \% FLOPs reduction with top-5 accuracy as 92.3\% (its baseline is 92.93\%).
>
> Q2: In the experiments on ImageNet, it takes too many epochs (180) to finetune the pruned models comparing with other methods (20 in SCOP [5]). It is unclear whether the good performance is resulting from longtime training.
>
> Thank you for the valuable comment. To prune ResNet-50 on ImageNet, SCOP first uses 20 epochs to optimize scaling factor $\beta^l$ and $\Tilde{\beta}^l$; and then it uses additional 120 epochs to fine-tune the model. We have re-conducted our experiment using the same 120 epochs for fine-tuning, and our pruned model still outperforms the model pruned by SCOP with respect to the accuracy, model size reduction and FLOPs reduction.
>
> Q3: To demonstrate the effectiveness of the proposed pruning metric, it would be better for the authors to show the accuracy-pruning rate trade-off curve of different methods, which will strengthen the paper.
>
> Thank you for the valuable suggestion. Following your comments, we study the accuracy-pruning rate trade-off for ResNet-50 on ImageNet. The results are shown below. The visualized curve can be seen in this link https://anonymous.4open.science/r/rebuttal-2C97/prune_ratio--acc_curve/varing_pruning_rate_f.png.
>
> Sparsity (\% )  [40.8, 44.2, 56.7, 68.6, 70.8, 72.5]
>
> Accuracy (\%) [76.30,76.15,75.26,73.3, 72.6, 72.3]
>
> FLOPs Reduction (\%) [44.8,48.7,62.8,76.7, 79.1, 80.9]
>
> Q4: Some important details are missing. How to set the pruning rate of each layer? Do the authors set the same pruning rate for each layer?
>
> Thank you for pointing it out! We use the similar setting of HRank for choosing the pruning rate of each layer. The details of the pruning rate for each layer are reported in this anonymous link https://anonymous.4open.science/r/rebuttal-2C97/kl%20and%20sparsity.md and the figure of sparsity in the "Sparsity" fold. As can be seen in this link, some of the layers are pruned at the same rate.  We will add that information to the supplementary materials of the final version of this paper.
>
> Q5: It would be better for the authors to conduct more experiments to investigate the effect of different N.
>
> Thank you for the valuable suggestion. Our experiments show that using different N has a very minor impact on performance. For pruning ResNet-56 on CIFAR-10, we calculate CI using N=640 and N=5120, respectively, and experimental results show that with the same 42.8\% parameter reduction, N=5120 setting brings accuracy as 94.12\%, and N=640 setting brings accuracy as 94.16\%. With the same 71.8\% parameter reduction, N=5120 setting brings accuracy as 92.03\%, and N=640 setting brings accuracy as 92.05\%. We will add the ablation study of using different N into the final version of this paper.
>
> Q6: It would be better for the authors to conduct more experiments on lightweight models (e.g., MobileNetV1 and MobileNetV2).}
>
> Thank you for the valuable suggestion. We are following your suggestion and conducting experiments on other CNN models. We will update the results in this discussion thread later.
>
> Q7: Please discuss the limitations of the proposed method.
>
> Thank you for the comment. The main limitation of this work is that, as an inter-channel correlation exploration approach, it needs more computational complexity than intra-channel method to determine the importance of filter.

---

> > ### Comment · Reviewer_TZ9P · 2021-08-26
> > **Feedback on rebuttal**
> >
> > I would like to thank the authors for their replies. After reading all the comments and the replies, the authors have solved most of my concerns. The only concern is that it would be better for the authors to show the accuracy-pruning rate trade-off curve of **different pruning metrics**, which will demonstrate the effectiveness of the proposed pruning metric. Therefore, I tend to keep my score.

---

> > > ### Author Response · Authors · 2021-08-26
> > > **Response to Reviewer4**
> > >
> > > Thank you for the valuable suggestion. The accuracy-pruning rate trade-off curve of different pruning metrics is reported in this anonymous link: https://anonymous.4open.science/r/rebuttal-2C97/prune_ratio--acc_curve/varing_pruning_rate_ff.png .
> > >
> > > Here the pruning metrics for CHIP, SCOP and HRank are nuclear norm-based channel independence, knockoff suppression and matrix rank, respectively. From the curve, it is seen that using the channel independence metric outperforms these existing pruning criteria.

---

> > > > ### Comment · Reviewer_TZ9P · 2021-08-28
> > > > **Feedback on rebuttal**
> > > >
> > > > I would like to thank the authors for their replies. After reading the response, I tend to raise my score.

---

> > > > > ### Author Response · Authors · 2021-08-31
> > > > > **Update on MobileNetV2 results**
> > > > >
> > > > > [Update on MobileNetV2 results]: Following the reviewer's suggestions, we perform the experiment of pruning MobileNetV2 model on ImageNet dataset via using CHIP method. The results and performance comparison are shown below. Please note that because HRank and SCOP do not prune MobileNetV2 on ImageNet, we compare our approach with the other three channel pruning approaches [R7-R9] that explicitly report this experiment.
> > > > >
> > > > > |  --  | Top-1 Accuracy(\%)  | FLOPs $\downarrow$(\%) |
> > > > > |  ----  | ----  | ---- |
> > > > > | ThiNet [R7]  | 63.75 | 44.7 |
> > > > > | DCP [R8]   | 64.22 | 44.7 |
> > > > > | DMC [R9]   | 68.37 | 46.0 |
> > > > > | CHIP       | 68.49 | 46.6 |
> > > > >
> > > > >
> > > > > From the experimental results, it is seen that our proposed CHIP outperforms these existing works when pruning MobileNetV2 model. Compared with the latest DMC method [R9], our approach achieves 0.12\% accuracy increase with even more FLOPs reduction.
> > > > >
> > > > >  [R7] Jian-Hao Luo, Hao Zhang, Hong-Yu Zhou, Chen-Wei Xie, Jianxin Wu, and Weiyao Lin. Thinet: pruning cnn filters for a thinner net. IEEE transactions on pattern analysis and machine intelligence (TPAMI), 41(10):2525–2538, 2018.
> > > > >
> > > > > [R8] Zhuangwei Zhuang, Mingkui Tan, Bohan Zhuang, Jing Liu, Yong Guo, Qingyao Wu, Junzhou Huang, and Jinhui Zhu.
> > > > > Discrimination-aware channel pruning for deep neural networks. In Advances in Neural Information Processing Systems (NeurIPS), pages 875–886, 2018.
> > > > >
> > > > > [R9] Shangqian Gao, Feihu Huang, Jian Pei, and Heng Huang.
> > > > > Discrete model compression with resource constraint for
> > > > > deep neural networks. In Proceedings of the IEEE/CVF Conference on Computer Vision and Pattern Recognition (CVPR), pages
> > > > > 1899–1908, 2020.

---

### Official Review · Reviewer_JzDy · 2021-07-10

**Rating:** 6
**Confidence:** 4

**Summary:**

This paper presents a metric to evaluate pre-trained dense networks to identify the feature maps that need to be dropped in an optimal manner, such that the performance of the pruned network is maximized. Authors present a simple, yet effective approach to identify the feature maps that need to be removed from the network such that it does not compromise the performance. Authors demonstrate the improvement on ImageNet dataset, which clearly demonstrates the efficacy of the proposed metric.

**Limitations And Societal Impact:**

Authors have not provided limitations of their work. I would recommend that the authors discuss limitations if any. Furthers, authors are recommended to add a line or 2 on the societal impact of their work - for example are there any assumptions made in designing the method that could lead to bias in any form that could adversely impact the society.

**Main Review:**

The paper appropriately addresses the scientific question of how to choose the correct set of feature maps for a pre-trained dense network and prune it. The idea of the paper is clear and the paper is well written. However, I have some comments/suggestions which need to be addressed during the rebuttal.

1. Authors state that the existing methods do not consider the inter-feature relations when choosing the feature maps to be eliminated during pruning. While I am not aware if any such method exists or not, it is important to note that most of the pruning method methods such that add additional objective (pruning-related) do actually consider this aspect in a implicit sense. For example, the “L1-regularization” approach increases sparsity at a layer level or at the entire network level. During this process, it takes an aggregate over a set of feature maps, thus tying them together. Similarly, there exist budget-aware pruning methods (“BAR” and “CHIPNET”) that introduce budget constraints in the pruning process, and this budget ties the different feature maps together during the selection process. My point here is that in such schemes, the mask corresponding to any feature map when goes towards a value of 0, it helps the other maps to go towards 1 in some sens – thus they are not really independent of each other. Authors seem to have completely missed this whole set of literature in their discussion. I would recommend the authors to accordingly soften their claim and add a discussion related to the above methods in the paper.

2. Authors claim that their one-shot static pruning approach prunes the networks in an optimal sense, and state that they have experimented with dynamic learning methods where further to-be-pruned feature maps are identified as a part of the learning process. However, in the supplementary material authors demonstrate this only in 3-4 lines with a figure that contains insufficient details. My personnel experience with static and dynamic pruning has has revealed that no matter how well the static pruning  is done, dynamic learning contrbutes significantly to the pruning results. It is certainly possible that the method proposed in the paper is better and its static nature is good enough, however, this needs to be rigorously evaluated before such a claim can be made. In this regard, I would like the authors to present thorough details for the extensive set of experiments that they have conducted to reach this conclusion.


**Time Spent Reviewing:**

2

---

> ### Author Response · Authors · 2021-08-10
> **Response to Reviewer3**
>
> We sincerely appreciate the reviewer's very constructive comments and suggestions. The following is our response in order of questions and comments raised.
>
> Q1: I would recommend the authors to accordingly soften their claim and add a discussion related to the above methods in the paper.
>
> Thank you for the valuable comments! We will change the tone and add the discussion related to the approaches indicated by the reviewer. We appreciate the reviewer for pointing them out very much.
>
> Q2: My personnel experience with static and dynamic pruning has revealed that no matter how well the static pruning is done, dynamic learning contributes significantly to the pruning results.
>
> Thank you for the valuable comment. We agree that conventional wisdom tells us that dynamic pruning should be able to bring better performance than one-shot static pruning, and that was the motivation why we explored to perform further dynamic pruning built upon the pruning mask identified by CHIP to improve performance, which we did not observe in our experiment. We hypothesize this phenomenon results from two reasons.
>
> First, the state-of-the-art static one-shot pruning approaches, including SCOP, HRank and our CHIP, trade considerable computation effort for better identification of the importance of filters. It is possible that the corresponding benefit after spending many computations might be that they can closely approach the optimal selection of filter masking, the benefit of using extra dynamic pruning seems to become relatively minor. And that might also explain why SCOP and HRank do not choose to adopt further dynamic pruning.
>
> Second, dynamic pruning, though powerful for unstructured pruning, seems to be not very efficient for structured pruning such as filter pruning. This is because, for filter-level dynamic pruning, the freedom of dynamic adjustment of pruning mask is quite limited, thereby largely affecting the performance. To date, most dynamic pruning works [R1][R2][R3] target unstructured sparsity, and to the best of our knowledge only DPF [R4] reports the performance on structured filter dynamic pruning, and it is inferior to our approach. To prune ResNet-56 on CIFAR-10 dataset, with 40\% pruning ratio, DPF suffers accuracy drop (94.51\%(baseline) 94.03\%(after pruning)); while our CHIP enjoys accuracy increase (93.26\%(baseline) 94.16\%(after pruning)) with 42.8\% pruning ratio. Notice that here our baseline accuracy is even 1.3\% lower than DPF's baseline.
>
> [R1] Guillaume Bellec, David Kappel, Wolfgang Maass, and Robert Legenstein. Deep rewiring: Training very sparse deep networks. In ICLR - International Conference on Learning Representations, 2018.
>
> [R2]Decebal Constantin Mocanu, Elena Mocanu, Peter Stone, Phuong H Nguyen, Madeleine Gibescu, and Antonio Liotta. Scalable training of artificial neural networks with adaptive sparse connectivity inspired by network science. Nature communications, 9(1):2383, 2018.
>
> [R3]Tim Dettmers and Luke Zettlemoyer. Sparse networks from scratch: Faster training without losing performance. arXiv preprint arXiv:1907.04840, 2019.
>
> [R4] Lin, Tao, et al. "Dynamic model pruning with feedback." arXiv preprint arXiv:2006.07253 (2020).
>
> Q3: I would like the authors to present thorough details for the extensive set of experiments that they have conducted to reach this conclusion.
>
> Thank you for the comment. To perform further dynamic learning upon the filter masks identified by Algorithm 1, we first initialize the score of \bm{M}_ {b_1,\cdots,b_m}^l based on the normalized CI. Starting from this initialization provided by CHIP, we then train the \bm{M}_{b_1,\cdots,b_m}^l using the MSE loss to minimize the gap between the Upper Bound of nuclear norm (see Fig.3) and current nuclear norm under given sparsity. We have tried many different combinations of hyperparameters for such further dynamic mask learning. For instance, for VGG-16 with 83.3\% sparsity, with optimizer of ADAM and SGD, we experiment different learning rates ranging from 0.1 to 0.001, different weight decays ranging from 0.05 to 5, and different epochs ranging from 100 to 1000. All of those results show such further dynamic mask learning procedure does not bring additional performance improvement, as compared to our one-shot filter pruning.
>
> Q4: I would recommend that the authors discuss limitations if any. Furthers, authors are recommended to add a line or 2 on the societal impact of their work.
>
> Thank you for the suggestion. The limitation of this work is that, because it explores inter-channel correlation, it needs more computational complexity than the intra-channel approach to determine the importance of filters. Regarding societal impact, a better filter pruning approach would promote the deployment of AI in a more energy-efficient way.

---

> > ### Author Response · Authors · 2021-08-26
> > **Update on dynamic pruning**
> >
> > [Update on dynamic pruning]: In our recent literature survey we find that the dynamic inference at the channel level is also called dynamic pruning in some papers [R5][R6]. The key idea of this strategy is to store the entire trained dense CNN model in the memory, and then dynamically remove/gate different channels according to different input in the inference phase. Compared with this input-aware strategy, our approach, together with other static pruning methods such as SCOP and HRank, have two benefits: 1) Static pruning can bring real memory cost reduction; while the input-aware dynamic pruning methods still need to store the entire dense models; and 2) for static pruning, the channel selection and removal are performed before the inference phase in an off-line way; while the input-aware dynamic pruning needs to calculate the importance of channels for each input during the inference, thereby causing extra computational overhead. In addition, by properly exploiting channel independence, our static pruning approach outperforms the recent input-aware dynamic pruning works. For instance, to prune ResNet-56 on CIFAR-10 dataset, our method can bring even 0.16\% accuracy increase (93.26\%(baseline) 93.42\%(after pruning)) with 59.6\% parameter reduction and 64.5\% FLOPs reduction; while [R5] has 0.06\% accuracy loss (93.7\%(baseline) 93.64\%(after pruning)) with 0\% parameter reduction and 62.4\% FLOPs reduction.
> >
> > [R5] Tang, Yehui, Yunhe Wang, Yixing Xu, Yiping Deng, Chao Xu, Dacheng Tao, and Chang Xu. "Manifold regularized dynamic network pruning." IEEE/CVF Conference on Computer Vision and Pattern Recognition (CVPR), pp. 5018-5028. 2021.
> >
> > [R6] Weizhe Hua, Yuan Zhou, Christopher M De Sa, Zhiru Zhang,
> > and G Edward Suh. Channel gating neural networks. In
> > Advances in Neural Information Processing Systems (NeurIPS), pages 1886–1896, 2019.

---

### Official Review · Reviewer_Z9Gi · 2021-07-15

**Rating:** 7
**Confidence:** 3

**Summary:**

To determine the importance of filters in DNNs, this paper proposes a new concept, channel independence, which measures the correlation of multiple feature maps. Also, this paper systematically investigates and analyzes the quantification metric, measuring scheme, and sensitiveness & reliability of channel independence in the context of filter pruning. As a result, the proposed scheme has achieved outstanding performance compared to existing methods. Although this paper is well-written and presents a solution for network pruning that has steadily gained a lot of attention, there are some minor comments as a result of my review of this paper. (Please refer to my main review)

**Limitations And Societal Impact:**

See my main review.

**Main Review:**

1. Scalability/Compatibility of the proposed method (Request for additional experiments)

1-1) Recently, activation functions with negative values such as Swish and Mish are widely used. In general, pruning is vulnerable to activation functions including negative values due to the influence of shift parameters, so please show whether the proposed method has robust properties to these activation functions.

1-2) As you know, fine-tuning requires a significant training cost, so many filter pruning techniques that achieve excellent performance without fine-tuning have been proposed. It is necessary to show whether good performance can be maintained even if fine-tuning is removed from the proposed method.

1-3) It would be good to present the results of various networks for ImageNet dataset such as ResNet-34/101 or MobileNet.

1-4) It is necessary to show whether the proposed method is effective not only for classification but also for more practical detection and segmentation networks.

1-5) Recently, there have been many attempts to perform dynamic pruning rather than static pruning within filter pruning, and therefore it would be good to include the comparison result and analysis with dynamic pruning schemes.

2. Request for additional explanation

2-1) Authors need to present an example in which the difference in the importance of filters occurs in the intra-channel and the inter-channel at the beginning of Chapter 1 or Chapter 3. And if possible, it would be good to present a numerical value indicating the proportion of such filters among all filters.

2-2) It seems that the complexity of including the inter-channel is relatively higher than considering only the intra-channel. Therefore, it is necessary to analyze the complexity.

2-3) Recently, many kernel pruning studies have been attempted, and therefore it would be good to explain the advantages of filter pruning considering inter-channel compared to kernel pruning targeting the kernel in the filter.

2-4) It seems that the baseline accuracy is slightly lower than the values originally presented in ResNet and other papers. Authors should explain how to obtain baseline accuracy.

**Time Spent Reviewing:**

4

---

> ### Author Response · Authors · 2021-08-10
> **Response to Reviewer2**
>
> We sincerely appreciate the reviewer's very constructive comments and suggestions. The following is our response in order of questions and comments raised.
>
> Q1: In general, pruning is vulnerable to activation functions including negative values due to the influence of shift parameters, so please show whether the proposed method has robust properties to these activation functions.
>
> Thank you for the valuable comment. Because our approach measures the channel independence from the inter-channel perspective, we believe it is robust to activation functions including negative values as well. Empirically, we conduct experiments on ResNet-56 with Swith activation. On CIFAR-10 dataset the baseline model achieves 93.28\% accuracy, and our pruned model with 42.8\% sparsity ratio achieves 93.22\% accuracy. This result shows the generality of our pruning approach across different activation functions. We will further perform more comprehensive experiments on other models and report the results in the final version of this paper.
>
> Q2: As you know, fine-tuning requires a significant training cost, so many filter pruning techniques that achieve excellent performance without fine-tuning have been proposed. It is necessary to show whether good performance can be maintained even if fine-tuning is removed from the proposed method. It is necessary to show whether good performance can be maintained even if fine-tuning is removed from the proposed method.
>
> Thank you for the valuable comment. Similar to HRank and SCOP, our filter pruning approach needs fine-tuning to achieve the desired performance. Although there are some fine-tuning-free pruning approaches recently, we believe fine-tuning-involved strategy (like ours and SCOP) is more competitive because of three reasons. First, the most important criteria to evaluate pruning performance is FLOPs reduction, model size reduction and accuracy; while the consumed time for pruning is not a prioritized factor. Second, many fine-tuning-free pruning approaches still needs costly procedures for pruning. For instance, as shown in Table 2, C-SGD needs extra training procedure for the pre-trained model to push filters identical to perform pruning. In such a scenario, it essentially replaces the post-pruning fine-tuning phase with the pre-pruning additional training, and hence such fine-tuning-free strategy does not really enjoy lower computing cost than fine-tuning-involved strategy. Third, using fine-tuning brings better accuracy and FLOPs reduction performance, which is the most important metric for pruning approaches. For instance, to prune ResNet-50 on ImageNet dataset, C-SGD suffers 0.29\% top-5 accuracy drop with 46.24\% FLOPs reduction; while our CHIP enjoys 0.04\% accuracy increase with 48.7\% FLOPS reduction.
>
> Q3: It would be good to present the results of various networks for ImageNet dataset such as ResNet-34/101 or MobileNet.}
>
> Thank you for the valuable suggestion. We are following your suggestion and conducting experiments on other CNN models. We will update the results in this discussion thread later.
>
> Q4: It is necessary to show whether the proposed method is effective not only for classification but also for more practical detection and segmentation networks.
>
> Thank you for the valuable comment. We are following your suggestion and conducting experiments on detection and segmentation networks. We will update the results in this discussion thread later.
>
> In addition, our original submission only conducts experiments on image classification is because of two reasons: 1) most of the state-of-the-art filter pruning works (e.g., SCOP, HRank) only report results on image classification dataset. For fair comparison, we follow the same task setting; and 2) image classification, as a fundamental computer vision task, is very general and representative to compare the performance of different pruning approaches.
>
> Q5: It would be good to include the comparison result and analysis with dynamic pruning schemes.
>
> Thank you for the valuable suggestion. Dynamic pruning is very powerful for unstructured pruning. However, we suspect it might not be very efficient for structured pruning like filter pruning. This is because the freedom of dynamic adjustment of pruning mask at the filter level is quite limited, thereby largely affecting the performance. To date, most dynamic pruning works [R1][R2][R3] target unstructured sparsity, and only DPF [R4] reports structured filter dynamic pruning. To prune ResNet-56 on CIFAR-10 dataset, with 40\% pruning ratio, DPF suffers accuracy drop (94.51\%(baseline) 94.03\%(after pruning)); while our CHIP enjoys accuracy increase (93.26\%(baseline) 94.16\%(after pruning)) with 42.8\% pruning ratio. In other words, with a higher pruning ratio and lower baseline model accuracy, our static one-shot filter pruning approach can still outperform dynamic filter pruning.
>
> [R1] Guillaume Bellec, David Kappel, Wolfgang Maass, and Robert Legenstein. Deep rewiring: Training very sparse deep networks. In ICLR - International Conference on Learning Representations, 2018.
>
> [R2]Decebal Constantin Mocanu, Elena Mocanu, Peter Stone, Phuong H Nguyen, Madeleine Gibescu, and Antonio Liotta. Scalable training of artificial neural networks with adaptive sparse connectivity inspired by network science. Nature communications, 9(1):2383, 2018.
>
> [R3]Tim Dettmers and Luke Zettlemoyer. Sparse networks from scratch: Faster training without losing performance. arXiv preprint arXiv:1907.04840, 2019.
>
> [R4] Lin, Tao, et al. "Dynamic model pruning with feedback." arXiv preprint arXiv:2006.07253 (2020).
>
>
> Q6: Authors need to present an example in which the difference in the importance of filters occurs in the intra-channel and the inter-channel at the beginning of Chapter 1 or Chapter 3. And if possible, it would be good to present a numerical value indicating the proportion of such filters among all filters.
>
> Thank you for your valuable comment.
> For the 21st layer of ResNet-50 model, among its 128 filters, when we want to remove the 50\% filters (64 filters), if using intra-channel(e.g., L1 norm)-based importance, then the indices of less important filters are (least important first):
>
> [ 91  18  41  14 108 102  24  33 114  27 117   3 115  51  10  82  52  11 35  37   9  65  12 113  85  39  23   5  61 103  19 107  21   8 121  46 86 125  95  69 124  73  98  90  34   4  48  87  89  43 122  78  29 116 13  71 123  44  15  84   0  70 111  76   7   6  47  25  83  45  94  50 72 109  60  30  64  38  88  40  53  58 106  56  17 110 119 104  59  92 2  66  22 120  36  68  75  16 100 101  31  81  93 105  99  97 112 80 32 127  74  79  49  20 126  63  55 118  67  62  28  57  42 1  26  54 96  77]
>
> On the other hand, when using our inter-channel criterion, the indices of filter importance are (least important first):
>
> [ 91  18  52  41 115  10   3  24  14  27  51 102   9  33 114  82  11 108 35 109 113  12  23  37 103 117  85  39 111  95  73 116  65   5  88  56 64 107 125  83  46 121  84  66  90 124  44  13  74  96  58   8  43  17 68 106  21  97  86  61  47  50  69  79  25   6  22  49   2   0 119  55 45   4  19  29  34 123   7  16  15  71 126  30  38  70  59  89  76  87 98  48 127 122  40  54  78  72 110  36  60  53  32  92  20  75  67 112 94 118  81 100  28  31 104  26 120  80  62 105   1  63  99 101  93  42 57  77]
>
> It is seen that for determining 64 less important filters, the intra-channel approach and inter-channel approach have different selections on 33.3\% filters.
>
> Q7: It seems that the complexity of including the inter-channel is relatively higher than considering only the intra-channel. Therefore, it is necessary to analyze the complexity.
>
> Thank you for the valuable suggestion. Given $c$ filters and corresponding $c$ feature maps, intra-channel filter pruning, such as HRank, calculates the importance (e.g., rank) of each size-$(h \times w)$ feature map for $c$ times. Our inter-channel approach calculates the nuclear norm of the combined flattened feature maps with $c$ times as well. And each combined flatten feature map is the size of $((c-1) \times hw)$. Though having higher complexity, the inter-channel approach can bring higher model complexity reduction and accuracy, which are the most important performance metrics for DNN pruning. We will add this complexity analysis to the final version of the paper.
>
> Q8: It would be good to explain the advantages of filter pruning considering inter-channel compared to kernel pruning targeting the kernel in the filter.
>
> Thank you for the valuable comment. Compared with kernel pruning, filter pruning enjoys the very important advantage of enabling practical speedup. More specifically, because filter pruning directly removes the entire filters (channels), the memory access in CPU/GPU is much more regular than that for kernel pruning. Therefore, filter pruning is a much more hardware-friendly solution that brings practical speedup on off-the-shelf computing devices.
>
> Q9: It seems that the baseline accuracy is slightly lower than the values originally presented in ResNet and other papers. Authors should explain how to obtain baseline accuracy.
>
> Thank you for the valuable comment. We obtain the pre-trained baseline models from Pytorch official model zoo. And the recent state-of-the-art channel pruning works (HRank and SCOP) also use the baseline models with the same accuracy. In addition, considering some other works use different baseline accuracy, in the experimental result section (Table 1 and 2) we report the change of accuracy (before and after using pruning) for a fair comparison.

---

> > ### Comment · Reviewer_Z9Gi · 2021-08-26
> > **Feedback on rebuttal**
> >
> > I would like to thank you for responding to my concerns. After carefully reading all of the other reviewers' comments and the author's feedback, I lean to keep my initial rating (7).

---

> > > ### Author Response · Authors · 2021-08-26
> > > **Update on dynamic pruning**
> > >
> > > We appreciate reviewer's feedback very much.
> > >
> > > We also have some updates on dynamic pruning. In our recent literature survey we find that the dynamic inference at the channel level is also called dynamic pruning in some papers [R5][R6]. The key idea of this strategy is to store the entire trained dense CNN model in the memory, and then dynamically remove/gate different channels according to different input in the inference phase. Compared with this input-aware strategy, our approach, together with other static pruning methods such as SCOP and HRank, have two benefits: 1) Static pruning can bring real memory cost reduction; while the input-aware dynamic pruning methods still need to store the entire dense models; and 2) for static pruning, the channel selection and removal are performed before the inference phase in an off-line way; while the input-aware dynamic pruning needs to calculate the importance of channels for each input during the inference, thereby causing extra computational overhead. In addition, by properly exploiting channel independence, our static pruning approach outperforms the recent input-aware dynamic pruning works. For instance, to prune ResNet-56 on CIFAR-10 dataset, our method can bring even 0.16\% accuracy increase (93.26\%(baseline) 93.42\%(after pruning)) with 59.6\% parameter reduction and 64.5\% FLOPs reduction; while [R5] has 0.06\% accuracy loss (93.7\%(baseline) 93.64\%(after pruning)) with 0\% parameter reduction and 62.4\% FLOPs reduction.
> > >
> > > [R5] Tang, Yehui, Yunhe Wang, Yixing Xu, Yiping Deng, Chao Xu, Dacheng Tao, and Chang Xu. "Manifold regularized dynamic network pruning." IEEE/CVF Conference on Computer Vision and Pattern Recognition (CVPR), pp. 5018-5028. 2021.
> > >
> > > [R6] Weizhe Hua, Yuan Zhou, Christopher M De Sa, Zhiru Zhang,
> > > and G Edward Suh. Channel gating neural networks. In
> > > Advances in Neural Information Processing Systems (NeurIPS), pages 1886–1896, 2019.

---

> > > > ### Author Response · Authors · 2021-08-31
> > > > **Update on MobileNetV2 results**
> > > >
> > > > [Update on MobileNetV2 results]: Following the reviewer's suggestions, we perform the experiment of pruning MobileNetV2 model on ImageNet dataset via using CHIP method. The results and performance comparison are shown below. Please note that because HRank and SCOP do not prune MobileNetV2 on ImageNet, we compare our approach with the other three channel pruning approaches [R7-R9] that explicitly report this experiment.
> > > >
> > > > |  --  | Top-1 Accuracy(\%)  | FLOPs $\downarrow$(\%) |
> > > > |  ----  | ----  | ---- |
> > > > | ThiNet [R7]  | 63.75 | 44.7 |
> > > > | DCP [R8]   | 64.22 | 44.7 |
> > > > | DMC [R9]   | 68.37 | 46.0 |
> > > > | CHIP       | 68.49 | 46.6 |
> > > >
> > > >
> > > > From the experimental results, it is seen that our proposed CHIP outperforms these existing works when pruning MobileNetV2 model. Compared with the latest DMC method [R9], our approach achieves 0.12\% accuracy increase with even more FLOPs reduction.
> > > >
> > > >  [R7] Jian-Hao Luo, Hao Zhang, Hong-Yu Zhou, Chen-Wei Xie, Jianxin Wu, and Weiyao Lin. Thinet: pruning cnn filters for a thinner net. IEEE transactions on pattern analysis and machine intelligence (TPAMI), 41(10):2525–2538, 2018.
> > > >
> > > > [R8] Zhuangwei Zhuang, Mingkui Tan, Bohan Zhuang, Jing Liu, Yong Guo, Qingyao Wu, Junzhou Huang, and Jinhui Zhu.
> > > > Discrimination-aware channel pruning for deep neural networks. In Advances in Neural Information Processing Systems (NeurIPS), pages 875–886, 2018.
> > > >
> > > > [R9] Shangqian Gao, Feihu Huang, Jian Pei, and Heng Huang.
> > > > Discrete model compression with resource constraint for
> > > > deep neural networks. In Proceedings of the IEEE/CVF Conference on Computer Vision and Pattern Recognition (CVPR), pages
> > > > 1899–1908, 2020.

---

### Official Review · Reviewer_nHQh · 2021-07-18

**Rating:** 5
**Confidence:** 5

**Summary:**

This paper proposes a channel pruning method named CHIP to improve the existing channel pruning frameworks. The proposed method measures the importance of channels through channel independence. Extensive experiments show the effectiveness of trunk pruning.

**Limitations And Societal Impact:**

The authors do not address the limitations and potential negative social impact of their work. The authors can discuss how the proposed pruning method affects the development of edge AI.

**Main Review:**

Pros.
(1)	The paper is well motivated and well-written. The contributions are clearly presented.

(2)	Model compression is an important topic in scaling deep models on mobile devices. It is essential to design an effective channel pruning method.

(3)	Extensive experiments are conducted to show the effectiveness of CHIP.

Cons.
(1)	The authors utilize the nuclear norm to measure the independence among channels. It is not clear for me to see the relationship between the nuclear norm and correlations between channels. Experiments should be done to show this.

(2)	The hyper-parameter $\mathcal{k}^l$ is crucial to determine the model size of the pruned model. But it is not specified in the paper. Note that the performance of the pruned model is strongly related to the layer-wise configuration of $\mathcal{k}^l$.

(3)	Since the motivation is to design the pruning method by exploring the correlation between channels. It would be great to visualize the correlation before and after pruning.

(4)	It would be better to see how CHIP performs on MobileNet. EfficientNet, etc.


**Time Spent Reviewing:**

12 hours

---

> ### Author Response · Authors · 2021-08-10
> **Response to Reviewer1**
>
> We sincerely appreciate the reviewer's very constructive comments and suggestions. The following is our response in order of questions and comments raised.
>
> Q1: It is not clear for me to see the relationship between the nuclear norm and correlations between channels. Experiments should be done to show this.
>
> Thank you for the valuable comment. The key idea of this paper is to measure the correlations among channels via calculating their linear independence from others. To achieve that, we first flatten the feature map of each channel to a vector and then combine all the vectorized feature maps to a huge matrix. Because rank mathematically represents
> the maximum number of linearly independent rows/columns of the matrix, the rank change, corresponding to before and after we remove one vectorized feature from the combined feature matrix, can then reveal whether that vectorized feature has strong correlations with others or not. However, as shown in Fig. 2, rank, as the l0-norm of singular values of a matrix, is sometimes too "hard" to reflect the change; while nuclear norm, as the l1-norm of singular values, is "softer" and can better measure such correlation more precisely.
>
> Q2: The hyper-parameter $k^l$ is crucial to determine the model size of the pruned model. But it is not specified in the paper.
>
> Thank you for pointing it out! Due to the space limit, in this response we only list the $k^l$ setting for sparsity 40.8\% ResNet-50 on ImageNet as follows:
> Conv layers from 1 to 49, the number of preserved filters are:
> [64, 41, 41, 230, 41, 41, 230, 41, 41, 230, 83, 83, 460, 83, 83, 460,83, 83, 460,83, 83, 460, 166, 166, 912, 166, 166, 912, 166, 166, 912, 166, 166, 912, 166, 166, 912, 166, 166, 912, 332, 332, 2048, 332, 332, 2048, 332, 332, 2048, 332, 332, 2048].
>
> All $k^l$ settings for our other pruned models are reported in this anonymous link https://anonymous.4open.science/r/rebuttal-2C97/kl%20and%20sparsity.md. And the figure format lies in the "Sparsity" fold.
> We will add that information to the supplementary materials of the final version of this paper.
>
> Q3: It would be great to visualize the correlation before and after pruning.
>
> Thank you for the valuable suggestion. The visualization for the correlation, which is measured as the nuclear norm, is illustrated in these two links. For ResNet-50, 21st layer:
>
> before pruning: https://anonymous.4open.science/r/rebuttal-2C97/visualize_CI/before_prune_CI.png
>
> after pruning 50\% filters: https://anonymous.4open.science/r/rebuttal-2C97/visualize_CI/after_prune_CI.png
>
> Q4: It would be better to see how CHIP performs on MobileNet. EfficientNet, etc.
>
> Thank you for the valuable suggestion. We are following your suggestion and conducting experiments on other CNN models. We will update the results in this discussion thread later.
>
> Q5: The authors can discuss how the proposed pruning method affects the development of edge AI.
>
> Thank you for the valuable suggestion. The proposed pruning method can bring more compact DNN models with high accuracy, thereby promoting the real-time energy-efficient deployment of AI at the edge. In particular, considering many embedded CPU/GPU platforms cannot support unstructured sparsity, the proposed high-performance structured pruning approach can produce hardware-friendly structured sparse DNN models that are very suitable in such a scenario.

---

> > ### Author Response · Authors · 2021-08-31
> > **Update on MobileNetV2 results**
> >
> > [Update on MobileNetV2 results]: Following the reviewer's suggestions, we perform the experiment of pruning MobileNetV2 model on ImageNet dataset via using CHIP method. The results and performance comparison are shown below. Please note that because HRank and SCOP do not prune MobileNetV2 on ImageNet, we compare our approach with the other three channel pruning approaches [R7-R9] that explicitly report this experiment.
> >
> > |  --  | Top-1 Accuracy(\%)  | FLOPs $\downarrow$(\%) |
> > |  ----  | ----  | ---- |
> > | ThiNet [R7]  | 63.75 | 44.7 |
> > | DCP [R8]   | 64.22 | 44.7 |
> > | DMC [R9]   | 68.37 | 46.0 |
> > | CHIP       | 68.49 | 46.6 |
> >
> >
> > From the experimental results, it is seen that our proposed CHIP outperforms these existing works when pruning MobileNetV2 model. Compared with the latest DMC method [R9], our approach achieves 0.12\% accuracy increase with even more FLOPs reduction.
> >
> >  [R7] Jian-Hao Luo, Hao Zhang, Hong-Yu Zhou, Chen-Wei Xie, Jianxin Wu, and Weiyao Lin. Thinet: pruning cnn filters for a thinner net. IEEE transactions on pattern analysis and machine intelligence (TPAMI), 41(10):2525–2538, 2018.
> >
> > [R8] Zhuangwei Zhuang, Mingkui Tan, Bohan Zhuang, Jing Liu, Yong Guo, Qingyao Wu, Junzhou Huang, and Jinhui Zhu.
> > Discrimination-aware channel pruning for deep neural networks. In Advances in Neural Information Processing Systems (NeurIPS), pages 875–886, 2018.
> >
> > [R9] Shangqian Gao, Feihu Huang, Jian Pei, and Heng Huang.
> > Discrete model compression with resource constraint for
> > deep neural networks. In Proceedings of the IEEE/CVF Conference on Computer Vision and Pattern Recognition (CVPR), pages
> > 1899–1908, 2020.

---

### Decision · Program_Chairs · 2021-09-27

**Decision:**

Accept (Poster)

**Comment:**

This paper proposes a new pruning method for model compression that utilizes channel independence (CI). CI is given by the change of nuclear norm of the entire set of feature maps between before and after removing a channel. The method is compared with other methods extensively, and it is reported that the proposed method shows a better performance than other existing ones.

This paper is overall well-written and it is easy to follow. The proposed method can be easily implemented. It is nice that such a simple method gives good performance improvement.
On the other hand, there are still some concerns as follows:
1. The authors state that the existing methods do not consider inter-channel correlation. However, it is a too strong claim and ignores several related works. For example, CCP [R1] and Spectral-Pruning [R2] consider correlation between channels, and other methods implicitly takes inter-channel correlation into account. The authors should add discussions about relation to those existing methods, and tone down the claim.
2. The proposed method utilizes a nuclear norm type criterion to measure channel independence instead of directly measuring channel correlations. The current form of the paper does not give sufficient justification about why the proposed one is appropriate. This point should be discussed properly, for example, by showing the strong correlation between channel independence (correlation score) and nuclear norm in numerical experiments.
3.  Some details of experimental settings such as the number of preserved filters $k_l$ are missing. These information should be included. I also recommend the authors to include accuracy-pruning rate trade-off curve of different pruning metrics.

[R1] Collaborative Channel Pruning for Deep Networks. ICML 2019.
[R2]  Spectral Pruning: Compressing Deep Neural Networks via Spectral Analysis and its Generalization Error. IJCAI 2020.

The concerns I listed above should be addressed. On the other hand, the proposed method shows fairly nice performance which is informative for the community. In summary, I recommend this paper for acceptance, while I think the authors should fix the issues listed above.